# Polysaccharide-Based Materials Created by Physical Processes: From Preparation to Biomedical Applications

**DOI:** 10.3390/pharmaceutics13050621

**Published:** 2021-04-27

**Authors:** Paulo R. Souza, Ariel C. de Oliveira, Bruno H. Vilsinski, Matt J. Kipper, Alessandro F. Martins

**Affiliations:** 1Group of Polymeric Materials and Composites, Department of Chemistry, State University of Maringá (UEM), Maringá 87020-900, PR, Brazil; pg53548@uem.br (P.R.S.); pg54114@uem.br (A.C.d.O.); bhvilsinski2@uem.br (B.H.V.); 2Laboratory of Materials, Macromolecules and Composites, Federal University of Technology—Paraná (UTFPR), Apucarana 86812-460, PR, Brazil; 3Department of Chemical and Biological Engineering, Colorado State University (CSU), Fort Collins, CO 80523, USA; 4School of Advanced Materials Discovery, Colorado State University (CSU), Fort Collins, CO 80523, USA; 5School of Biomedical Engineering, Colorado State University (CSU), Fort Collins, CO 80523, USA

**Keywords:** growth factors, scaffolds, antimicrobial coatings

## Abstract

Polysaccharide-based materials created by physical processes have received considerable attention for biomedical applications. These structures are often made by associating charged polyelectrolytes in aqueous solutions, avoiding toxic chemistries (crosslinking agents). We review the principal polysaccharides (glycosaminoglycans, marine polysaccharides, and derivatives) containing ionizable groups in their structures and cellulose (neutral polysaccharide). Physical materials with high stability in aqueous media can be developed depending on the selected strategy. We review strategies, including coacervation, ionotropic gelation, electrospinning, layer-by-layer coating, gelation of polymer blends, solvent evaporation, and freezing–thawing methods, that create polysaccharide-based assemblies via in situ (one-step) methods for biomedical applications. We focus on materials used for growth factor (GFs) delivery, scaffolds, antimicrobial coatings, and wound dressings.

## 1. Introduction

Polysaccharides have hydrophilic functional groups (charged groups, as well as hydrogen bond donors and acceptors) that can stabilize macromolecular assemblies. Polysaccharide assembly can also be achieved via electrostatic crosslinking using small-molecule or metal counterions, and through cooling and freezing–thawing of polysaccharide-based mixtures. These assemblies include polyelectrolyte complexes (PECs), polyelectrolyte multilayers (PEMs), coacervates, and hydrogels. PECs are assemblies mainly formed from the electrostatic complexation in solution of oppositely charged polyelectrolytes. The resulting complexes may remain highly hydrated, and are therefore often characterized as hydrogels or coacervates as well. Coacervates are the result of a liquid–liquid phase separation, resulting in a polysaccharide-rich (liquid) phase that remains hydrated and is suspended in an (aqueous) solution. Coacervates and PECs often have polydisperse size distributions. Hydrogels are hydrophilic condensed (solid) networks of macromolecules, which are capable of absorbing large amounts of water (greater than 90% by weight). Whether formed by coacervation or gelation, the result is often three-dimensionally structured nano- or microparticles. Polysaccharides can also be assembled through various film-forming, fiber-spinning, and phase-separation methods. Films are often obtained by solvent evaporation method or through the layer-by-layer assembly of PEMs [1,2,3]. The formation of polysaccharide PECs, coacervates, hydrogels, fibers, and films is generally achieved at relatively mild conditions; however, processing conditions used for any of these assembly methods (e.g., solution pH, concentration, temperature, and ionic strength) can greatly influence the resulting material structure and properties.

Polysaccharide-based materials have been used as wound dressings, drug delivery systems (DDSs), scaffolds, and coatings for tissue-engineering purposes [4,5]. Polysaccharides are attractive materials for these applications due to their cytocompatibility, biodegradability, high bioavailability, and natural abundance [5]. Many polysaccharides also exhibit antimicrobial, antimycotic, anti-adhesive, anticoagulant or procoagulant, and wound-healing properties. They have hydrophilic groups (carboxylic acids, amino, hydroxyl, and sulfate groups) in their structures that support bio-adhesion through non-covalent bonds toward biological tissues and growth factors (GFs) [6]. Some polysaccharides naturally occur in the extracellular matrix, and play important roles in binding proteins, cells, and tissues.

This review summarizes recent advances in developing polysaccharide-based materials for biomedical materials. Section 1 introduces the principal polysaccharides used in biomedical-engineering applications; Section 2 presents the main strategies used to create physically associated polysaccharide-based materials for medical applications; Section 3 discusses particular applications organized around the types of formulations based on polysaccharides. We focus on polysaccharide-based scaffolds, wound dressings, and DDSs for GFs, discussing their characteristics that make their controlled delivery challenging. We then discuss how polysaccharide-based materials are ideally suited to overcome the most important challenges, with discussion of some disadvantages as well.

## 2. Principal Polysaccharides Used for Biomedical Materials

Polysaccharides can be chemically stable, pH-responsive, and thermosensitive. These properties, combined with their chemical and biochemical functionality, gelling properties, and structural similarity to extracellular matrix components make them excellent candidate materials for use in biological systems. Here, we highlight the properties of glycosaminoglycans (GAGs) [7], alginate [8], chitosan [9], carrageenans, ulvan, fucoidan [7], and polysaccharide derivatives (especially sulfated materials [10]). Polyanionic polysaccharides (GAGs and marine polysaccharides) have often been used to develop DDSs for cationic GFs [11] and surface coatings. Cationic polysaccharides (chitosan and its derivatives) comprise DDSs for anionic GFs [12], surface coatings, wound dressings, and scaffolds with antimicrobial properties. Our principal focus is on charged polyelectrolytes (polyanionic and polycationic polysaccharides) because these can mainly interact through electrostatic interactions, forming durable assemblies (physical materials) for biomedical applications. Moreover, we focus on cellulose (a neutral polysaccharide), because it is the most abundant polysaccharide in the world. It provides nanocrystalline structures that improve the mechanical properties of polysaccharide-based materials, and bacterial cellulose is attracting significant attention for biomedical applications.

### 2.1. Glycosaminoglycans (GAGs)

GAGs are linear anionic polysaccharides mainly composed of disaccharide units containing a hexuronic acid (glucuronic acid or iduronic acid) and a hexosamine (glucosamine, or galactosamine). GAGs comprise complicated chemical structures, distinguished by their specific disaccharide repeat sequences, glycosidic bonds, and substituents (*O*-sulfates, *N*-sulfonates, and *N*-acetyl groups). They are present in many human and animal tissues, and are obtained commercially from the tissues of pigs, poultry, sharks, and reptiles. GAGs molecular masses mainly depend on the extraction method and source. They include sulfated polymers, such as heparin, heparan sulfate, chondroitin sulfate, dermatan sulfate, and keratan sulfate [7,13,14] (Figure 1). Hyaluronic acid (often called hyaluronan) is the only non-sulfated GAG.

The most important GAGs for biomedical applications are heparin, chondroitin sulfate, and hyaluronic acid, because they are abundant extracellular membrane components. Heparin has a linear chain consisting of an alternating sulfated uronic acid and d-glucosamine units linked by α- and β bonds (1→4). The uronic acid can be l-iduronic or d-glucuronic acid, while the d-glucosamine is *N*-sulfated or *N*-acetylated. The l-iduronic acid is sulfated at the C2 position, and the d-glucosamine unit is *N*- and 6-*O* sulfated. l-Iduronic acid corresponds to approximately 85% of the uronic acid content, and d-glucuronic acid comprises 15% [7,16].

Chondroitin sulfate is composed of repeating β-1,3-linked *N*-acetyl galactosamine and β-1,4-linked d-glucuronic acid disaccharide units [14,17]. The chemical structure depends on the sulfate groups’ positions on the pyranose ring and sulfation degree. It is often classified as chondroitin sulfate A, C, D, and E (Figure 1). Hyaluronic acid has the highest molecular mass among the GAGs, and it is composed of β-1,4-d-glucuronic acid and β-1,3-*N*-acetyl-d-glucosamine disaccharide units (Figure 1) [18]. The high molecular mass of hyaluronan imparts viscoelastic and bio-adhesive properties to materials [19,20]. It is a major component of the extracellular matrix of many tissues, including skin [21].

Sulfated disaccharides on GAGs containing sulfate and carboxylic groups have pK_a_ values between 2.0 and 4.0 [16,22,23]; therefore, all the sulfated GAGs are ionized in water and biological fluids. The ionized sulfates are hydrophilic, making them water-soluble. Sulfated GAGs can strongly bind positively charged proteins [24]. The physicochemical and biochemical properties of GAGs (hydrophilicity, biocompatibility, biodegradability, and chemical cues that regulate major biological processes, including cell growth and differentiation) rely on their chemical structure, specific architecture, sulfation degree, molecular mass, and conformation in solution [21,25,26]. Low-molecular-weight heparin has anti-inflammatory properties and anticoagulant effects, whereas unfractionated heparin has less predictable and controllable biological properties [27].

The sulfated GAGs occur covalently end-grafted to proteins, forming three-dimensional bottlebrush structures called proteoglycans [28]. Proteoglycans are complex macromolecules found in cell membranes, the extracellular milieu, and intracellular granules. These structures can have different amino acid sequences, lengths, and different types and numbers of GAGs attached to their backbones [29]. Proteoglycans and their constituent GAGs are responsible for many biochemical functions of the extracellular matrix and cell membranes, including organizing the nano- and microstructure; enhancing tribological and mechanical properties; regulating the transport of oxygen and nutrients; restoring the structure and function of damaged tissues; providing microenvironments for cell survival [30]; and binding, stabilizing, and activating GFs to control signaling [31].

### 2.2. Chitin and Chitosan

Chitin is a linear polysaccharide mainly composed of β (1→4) units linked to *N*-acetyl-2-amino-2-deoxy-d-glucose residues found in fungi cell walls (*Aspergillus niger*, *Penicillium chrysogenum*, *Penicillium notatum*, and others), and in the exoskeletons of crustaceous (shrimps, lobster, krill, goose barnacle, and crabs), insects (cockroach, ladybird, butterfly, and others), algae (Phaeophyceae, Chlorophyceae, and others), and mollusks (cuttlefish, octopus, and squids) [32,33]. It occurs in different polymorphic forms (α, β, and γ-chitin). The α-chitin arranged in anti-parallel strands is the most stable and abundant form [34]. Chitin is biodegradable and mainly extracted from crustacean wastes that are byproducts of the food fishing industries, comprising an acetylated polymer with aqueous insolubility [33,35].

Chitin can be formulated into materials (films, beads, hydrogels, and fibers). However, these materials are mainly prepared in volatile organic solvents, ionic liquids, and NaOH/urea mixtures [36]. Residual traces of these solvents are potentially toxic for biomedical applications. Chitin deacetylation in aqueous alkaline solutions at 60–80 °C creates the partially acetylated chitin derivatives [32,35]. Deacetylation degrees higher than 50% are referred to as chitosan, which are random copolymers of *N*-acetyl d-glucosamine and 2-amino-2-deoxy-β-d-glucosamine residues (Figure 2).

Chitosan is a linear cationic polysaccharide with pendent amine groups. The amine moieties on chitosan are protonated at low pH, making chitosan soluble in dilute acidic aqueous solutions [37]. Chitosan advantages over chitin include enzymatic degradation [38], gelling properties [39,40], pH-responsiveness [41], mucoadhesion, ability to open epithelial tight junctions (due to its cationic behavior that enhances interactions with mucous membrane [42]), and antimicrobial activities [43]. Molar mass and acetylation degree significantly influence the processing of chitosan-based materials [32]. These properties affect the chitosan hydrophobicity, solubility, viscosity, rheological, and gelling features. Chitosan gelation temperature decreases from 75 to 30 and 25 °C when the deacetylation degree is 83, 94, and 96%, respectively. Higher loss modulus (G’’) indicates that the gelation of chitosan solutions forms weak structures, and high deacetylation degrees support stiffer networks due to the effective H-bonds and polymer entanglements [44].

The protonated amino groups on chitosan interact with anionic materials at suitable pH [45]. Chitosan complexes with anions and polyanions have been used to encapsulate proteins. Their pH responsiveness can be used to modulate the release of proteins while protecting them against degradation [46,47,48,49]. The anionic materials commonly used to form complexes with chitosan include alginate [50,51], collagen [52,53], gelatin [9,54,55,56], poly(γ-glutamic acid) [57], β-glycerophosphate [58,59,60], and tripolyphosphate [47,61]. Chitosan can associate with synthetic polymers (poly(vinyl alcohol) [56,62], polyethylene glycol [63], and poly(lactic-*co*-glycolic acid) [64]) and other materials (including clays [65] and graphene oxide [52]) for producing DDSs with enhanced mechanical properties and hydrophilic–hydrophobic balance.

Chitosan generally has low solubility in biological fluids. It can easily be modified to overcome this disadvantage. Chitosan has a reactive amino group at carbon C2 and hydroxyl groups at carbons C3 and C6, in the deacetylated residues. Many reports of chemical modification of chitosan have been published. These chitosan derivatives include graphitized copolymers with poly(ε-caprolactone) [66,67], and polyethylene glycol [67,68,69], carboxymethyl chitosan [70,71,72], *N*-succinyl-chitosan [73,74], hydroxyphenyl acetamide chitosan [73], *N*,*N*,*N*-trimethyl chitosan [75], and chitosan conjugated with mesoporous silica nanoparticles [12,76] (Figure 3). These chitosan-based materials have been used in biomedical applications.

One of the most reported types of chitosan derivatives is carboxymethyl chitosans. These can be prepared from different synthetic pathways [77]. *O*-Carboxymethyl chitosan is prepared by suspending chitosan in an isopropanol/NaOH mixture and dropping (slowly) monochloroacetic acid in isopropanol in the suspended chitosan at 55 °C. This synthesis occurs using a NaOH excess to prevent the *N*-carboxymethylation. *N*-Carboxymethyl chitosan is obtained through the reaction between the free amines on chitosan with glyoxylic acid and sodium borohydride at pH between 3.2 and 4 (60 °C). *N*,*O*-Carboxymethyl chitosan is prepared by dissolving chitosan in an isopropanol/sodium hydroxide/chloroacetic acid mixture in a low NaOH concentration at 50 °C. *N*,*N*-dicarboxymethyl chitosan is prepared by tuning the chitosan, water, acetic acid, glyoxylic acid, and sodium borohydride contents at pH between 2 and 3. The ratio between amine and glyoxylic moieties should be 1:9. These chitosan derivatives are also used in biomedical applications [77].

### 2.3. Alginates

Alginates are natural polyuronates that have been used to engineer injectable drug delivery devices because of their low-cost of production, cytocompatibility, gelling, mucoadhesive, and pH-responsive properties [78,79,80,81,82]. Alginates are marine polysaccharides and comprise linear anionic polymers extracted from brown algae (*Phaeophyceae*, including *Laminaria hyperborean*, *Laminaria japonica*, *Laminaria digitata*, *Ascophyllum nodosum*, and *Macrocystis pyrifera*) [83]. Alginate hydrogels established by divalent cations (magnesium, calcium, barium, and strontium) naturally occur in the *Phaeophyceae* extracellular matrix. The seawater equilibrium influences the counter-ion types found in alginates [84].

The alginate repeat units are composed of (1,4)-linked β-d-mannuronic acid (M) and α-l-guluronic acid (G) residues. The structure of alginate is characterized by homopolymer blocks (MMMM or GGGG) or alternating copolymer blocks (MGMG) [85] (Figure 4A). The relative percentages of M, G, and MG blocks depend upon the seaweed algae source and extraction method, that generally involves (i) acid extraction, (ii) filtration and washing steps, and (iii) filtrate solubilization in an aqueous NaOH solution to create sodium alginate. Further steps of floatation, centrifugation, filtration (to remove impurities and insoluble particles), precipitation (in alcohol), and extraction with barium ions are also carried out. Barium ions have a high affinity to bind to the anionic moieties on alginates, separating them from cytotoxic impurities. Alginates are recovered by precipitation, forming sodium alginates for biomedical materials [83,84]. These procedures provide purified and water-soluble alginates that are stable alginate gels in mildly acidic conditions [86]. On the other hand, they have instability in alkaline medium. The water solubility depends on the pH, ionic strength and presence of metallic cations in the aqueous solutions [85,87].

Higher amounts of G blocks provide stiff alginate-based materials due to axial links and desirable chain conformation to form well-established egg-box structures with metallic cations (especially with calcium ions, Figure 4B) [80,89]. Both G and GM blocks participate in the egg-box gelation mechanism with divalent cations. High contents of alternating GM sequences increase the aqueous alginate solubility, while the gelation features mainly rely on the alginate molecular mass, G/M ratio, and pH. A higher molecular mass improves the gelling properties of aqueous alginate solutions due to the increase of polymer viscosity, supporting polymer entanglements, and thereby elastic and durable alginate-based materials [90,91,92]. The G/M ratio plays an essential role in the gelation process. At low pH (pH < 2.5), alginates are protonated, making water-insoluble alginic acids [85].

### 2.4. Other Marine Polysaccharides

Sulfated polysaccharides (composed of iduronic and glucuronic acids, galactose, fucose, and rhamnose) identified in algae have attracted significant attention for biomedical applications [93,94,95,96] because they resemble GAG structures [95]. These water-soluble anionic polymers include carrageenans (isolated from red Rhodophyceae), fucoidan (extracted from brown Phaeophyceae), and ulvan (obtained from green Ulvales and Chlorophyta) [93,97] (Figure 5). These polymers are abundant, cytocompatible, biodegradable, and present immunogenic, anti-inflammatory, anticoagulant, and gelling properties [93,98,99]. These sulfated marine polysaccharides can be used as a cheap feedstock for replacing GAGs in biomedical applications [93,100].

Carrageenans are linear polysaccharides often classified as *kappa* (*κ*), *iota* (*ι*), and *lambda* (*λ*) carrageenans according to the sulfation degree [97]. Carrageenans have thermo-reversible gelling (principally at the presence of potassium and calcium ions), viscoelastic, thickening, and stabilizing properties [102]. They are formed by alternating α-(1→3)-d-galactose-4-sulfated and β-(1→4)-3,6-anhydro-d-galactose units; *κ-*carrageenan has one sulfate per disaccharide, while *ι-*carrageenan has two sulfates, and *λ-*carrageenan has three sulfates (Figure 5).

Fucoidan naturally occurs in the cell walls of brown seaweeds with high contents of l-fucose and sulfate ester sites [103,104,105]. However, fucoidan (heteropolysaccharide) also contains α (1→3) l-fucopyranose with alternating α (1→3) and α (1→4)-linked l-fucopyranosyls sulfated at the C2 and C4 positions. These can occur acetylated with side branches (fucopyranoses or glucuronic acid) as well [103]. The most common fucoidan structures are the structures type I and type II (Figure 5) [101]. Glucose, xylose, galactose, and mannose monosaccharides can also comprise the fucoidan structure [104,106,107,108,109]. Fucoidan can resemble the heparin chain depending on the content of sulfate groups in its repeat unit (Figure 5). Ulvan comprises a branched anionic polysaccharide composed of sulfated rhamnose, iduronic, and glucuronic acids [110]. Its chemical structure is similar to GAGs, containing glucuronic acids and sulfated moieties [93,111].

### 2.5. Cellulose

Cellulose is a linear crystalline polysaccharide composed of β-d-glucopyranose units linked by β-1,4-glycosidic bonds (Figure 6A). It is the most abundant naturally derived polysaccharide found on Earth. The effective intramolecular H-bonding between its linear chains prevent mobility and impart aqueous insolubility [112]. Cellulose occurs in plants, including walnut shells [113], *Ampelodesmos mauritanicus* [114], shaddock peel [115], corn cob [116], and many other sources [117]. It comprises semi-ordered structures called microfibrils (Figure 6A). Cellulose microfibrils naturally occur in microbial biofilms [118,119,120], marine animals (e.g., *Halocynthia roretzi* [121]), and algae (green, gray, red, and yellow-green [112]). Cellulose microfibrils support structural reinforcement in biomaterials.

In plants, cellulose naturally occurs in nanoscale domains with hemicellulose, lignin (phenolic materials), waxes, trace elements, and impurities. Cellulose properties depend upon how it is obtained from lignocellulosic sources (including chemical, mechanical, biological, enzymatic, and combinations of these approaches). Chemical extraction uses toxic compounds (sodium hydroxide, sodium disulfide, chlorine dioxide, hydrogen peroxide, and peracetic acid). Chemical processes provide lignin by-products that can preclude the use of cellulose in biomedical applications. Mechanical processes (micro fluidization, cryo-crushing, ultrasonication, and others) have a high energy demand. Bacteria and enzymes bind hemicellulose, thus disrupting the lignin and cellulose interface and removing lignin-associated hemicellulose. This is a straightforward method that reduces cellulose degradation [112].

The extraction from algae and animals has some disadvantages as well. From animals (*Halocynthia roretzi*), the isolation often involves (i) hydrolysis (sulfuric acid, 180 °C, 2 h), (ii) kraft cooking step carried out in aqueous sodium hydroxide and sodium disulfide (180 °C for 2 h) to eliminate proteins and sugars with washing and drying, and (iii) bleaching in an aqueous sodium hypochlorite solution (75 °C for 1 h). These methods do not significantly damage the crystalline cellulose domains. Methods of extraction from algae include washing steps with water, Soxhlet extraction in a binary toluene/ethanol mixture, alkalization with sodium hydroxide (80 °C for 2 h), and bleaching with sodium chlorite and hydrogen peroxide [112]. The main disadvantages of these methods include the time required, the low extraction efficiency, and the use of organic solvents that increase the production cost, requiring additional purification steps [128].

Purified cellulose with a high crystalline microfibrillar structure is synthesized by bacteria (e.g., *Komagataeibacter xylinus* and *Gluconacetobacter hansenii* [112,118]. A dilute sodium hydroxide solution removes impurities, and after washing and drying steps, cellulose membranes are easily formed (Figure 6C) [118]. The advantage of this process is that purified or hydrated cellulose is created without complicated processes using harsh chemicals (Figure 6C). These features are critical to developing safe cellulose-based materials for biomedical applications.

Nanocellulose-based materials (nanofibrils and rodlike cellulose microcrystals, commonly called nanowhiskers, Figure 6B) have attracted considerable attention. These nanomaterials have a high surface area-to-volume ratio, renewability, optical transparency, surface functionality available for chemical modifications (containing primary and secondary hydroxyl groups at C6 and C2–C3, respectively), crystallinity, outstanding mechanical properties, cytocompatibility, and biodegradability [124,129,130,131,132]. They comprise ordered structures with packed parallel cellulose networks maintained by van der Waals and effective H-bonding interactions [129]. These nanomaterials are also isolated from plants, marine animals, algae, bacteria, and paper waste, following acid and enzymatic hydrolysis, mechanical and oxidation methods, strategies with ionic liquids, subcritical water hydrolysis, and associated processes [112,130,133]. Acid hydrolysis is the most common preparation method (Figure 6B) [129].

Cellulose-based derivatives such as methylcellulose, thiolated cellulose [126], ethylcellulose [125], hydroxypropyl methylcellulose [134], cellulose acetate, and others [135] have been used in biomedical applications. Figure 6D shows the more common pathways used to synthesize these cellulose derivatives. Many of these cellulose derivatives have better solubility than cellulose in organic and aqueous media, and are used as precursors to prepare more complicated cellulose derivatives. For example thiolated cellulose has a higher affinity toward proteins, due to the presence of thiol binders [125]. Methyl and ethyl celluloses are hydrophobic, while both acetate and hydroxypropyl cellulose are water-soluble [127]. These derivatives can be used in the design of smart delivery architectures, including films, electrospun fibers, and coatings, for biomedical applications.

### 2.6. Artificially Sulfated Polysaccharides

Hyaluronic acid, alginate, and chitosan have been chemically modified to provide sulfated derivatives with structures similar to heparin. These derivatives (often called heparinized materials) mimic some features and functions of heparin. Figure 7 shows the most common pathways used to synthesize artificially sulfated polysaccharides. The disadvantage of these processes is that chemical procedures significantly decrease the molecular mass of the native polysaccharide, due to the severely acidic conditions required (using chlorosulfonic and sulfuric acids) (Figure 7). The polysaccharide hydrolysis provides materials with high polydispersity index and low molar masses [136].

## 3. Processing Polysaccharide-Based Materials for Biomedical Applications

The polysaccharides reported in Section 2 can be used to engineer many physical materials, including nano- and microparticles, porous hydrogels, films, fibers, coatings, membranes, etc. This section highlights the principal strategies used to prepare these polysaccharide-based materials. We discuss layer-by-layer assembly, electrospinning, coacervation, ionotropic gelation, freezing–thawing, solvent evaporation (“casting”), gelation of polymer mixtures, and tridimensional bioprinted approaches (Figure 8). These strategies often yield physical assemblies based on polysaccharides for biomedical applications following in situ (one step) methods.

The versatile processability of some polysaccharides means that diverse materials (such as beads, films, fibers, porous hydrogels, particles, etc.) using the same polysaccharide system. This broad repertoire of materials requires developing appropriate methods and processing conditions to design different structures (Figure 8). For example, chitosan/alginate beads are produced by dropping aqueous alginate solution into chitosan (pH 1.0) containing sodium chloride. Bead formation can be controlled by tuning the ionic strength of the chitosan solution [87]. Aqueous chitosan and pectin solutions create porous hydrogels by cooling chitosan/pectin blends (60 °C), followed by lyophilization [137]. However, the solvent evaporation and layer-by-layer methods produce films and thin films by associating aqueous chitosan and pectin solutions, respectively [37,138]. Other adjustable processing parameters include polymer concentration, ionic strength, pH, temperature, and drying process. All of these processing parameters can affect the intermolecular forces that govern polysaccharide assembly, by electrostatic, hydrogen bond, and hydrophobic interactions, as well as polymer chain entanglements. The resulting physical materials can maintain their structures in aqueous media, making them suitable for use in biomedical applications (Figure 8). Moreover, polysaccharide-based materials generally have high cytocompatibility and biodegradability. Many polysaccharides are degraded by enzymes, resulting in non-toxic degradation products.

### 3.1. Thin Films and Coatings

Methods for depositing ultrathin (nanometer-scale) films and coatings on solid substrates are widely used in many applications [149], especially to create antimicrobial, anti-adhesive, and anti-fouling surfaces, drug delivery systems for GFs [46], and coated scaffolds [150]. The layer-by-layer (LbL) approach is a straightforward, versatile, and inexpensive method that provides thin films and coatings on solid substrates [151]. It is a widespread method for coating solid substrates with polysaccharide-based PEMs. The conventional LbL method based on the dipping strategy was first reported by Iler et al. (1966) and improved by Decher et al. (1991). Since then, it has attracted enormous interest, especially in the last two decades [152]. Other approaches are reported to create PEMs on solid substrates, including spin-coating, spraying, and combined spin–spraying methods [149].

The LbL method is based on the deposition/adsorption of PEMs on a solid substrates [37,149]. The solid substrate should be modified to interact with and adsorb charged polymers on its surface. Suitable solid substrates, including metals, ceramics, glasses, metal oxides, and polymers, are often oxidized by using chemical methods, oxygen gas plasma, and ultraviolet light combined with ozone. Then, the oxidized substrate can adsorb polycationic polymers. The LbL approach produces thin films and coatings with controlled thicknesses (often between 1 and 100 nm) by controlling the number of layers deposited on the solid substrate [153,154,155,156,157].

PEMs are mainly established by electrostatic interactions between charged polyelectrolytes, while hydrophobic and H-bonding interactions between the layers stabilize the PEMs [158]. Chitosan is the main polycationic polysaccharide used to build up PEMs [46]. It is firstly adsorbed on oxidized substrates, followed by a polyanionic layer (e.g., heparin, chondroitin sulfate [46], hyaluronic acid [140], and others). The number of alternated polycationic and polyanionic layers can be adjusted to control the thickness [153]. The substrate surface can be partially or entirely covered by PEMs depending on the number of layers adsorbed or deposited. This enables fine control over the surface physicochemical properties (wettability and roughness) of nanocoatings.

### 3.2. Polysaccharide-Based Precipitates and Coacervates

Polysaccharide-based precipitates can be quickly and easily prepared by combining an aqueous solution of a polycation (e.g., chitosan and proteins [159,160]) with an aqueous solution containing a polyanion (e.g., glycosaminoglycans, alginate, *κ*-carrageenan [161], and gums [162]). This strategy can prepare nanoparticles, microparticles [163], beads [87], and porous hydrogels, following a one-step method in situ. Polysaccharide-based complexes are mainly used as DDSs and scaffold matrices. Key parameters influence the precipitate and coacervate preparation, including the polyelectrolyte types, polymer molecular mass, polymer concentration, pH, temperature, and ionic strength [160,164,165,166]. These parameters influence the precipitate formation; however, the mixture of concentrated solutions often provides precipitate. These parameters should be tuned to provide a high complexation yield [167]. 

Precipitate and coacervate structures are stabilized by intermolecular interactions (Coulomb, ion–dipole, H-bonds, dipole–dipole, and hydrophobic forces) established between polyelectrolyte pairs [168]. The complexation between chitosan and anionic polysaccharides provide ΔH° < 0 and ΔG° < 0. Sulfated polyanions (e.g., *κ*-carrageenan) interact better with chitosan (performing irreversible precipitates than carboxylates polyanions (e.g., gums), which often form reversible polyelectrolyte complexes [161]. 

Nanoparticles in aqueous suspension (coacervates) are created by mixing polyelectrolyte solutions of low concentration, with one of the two polyelectrolytes in stoichiometric excess, to avoid the formation of large precipitates. The nanoparticle features, including size, surface charge (Zeta potential), water uptake capacity, and durability, can be adjusted using an appropriate ratio of polycation to polyanion balance [169]. When one polyelectrolyte is in excess, the resulting nanoparticles formed from the complexation of polyanion and polycation can contain a hydrophobic core, surrounded by an excess of one of the two polyelectrolytes near the surface, which imparts a high surface charge density to the particles. The resulting electrostatic repulsion between particles stabilizes the nanoparticles against aggregation in solution, and limits the size of the particles as they form. The one-shot addition of chitosan and glycosaminoglycans (heparin, chondroitin sulfate, and hyaluronic acid with concentrations between 0.9 and 1.9 mg/mL) solutions in a desirable chitosan/glycosaminoglycan volume ratio (approximately 20/80 *v*/*v*) provides nanoparticles with a hydrodynamic radius ranging from 110 to 219 nm and Zeta potential between −24 and −41 mV, respectively [170]. The excess of polyanionic materials provides nanoparticles with negative Zeta potentials. These nanoparticles were prepared by mixing individual polyelectrolytes solutions in an acetic acid/acetate buffer (0.1 M) at pHs 4.6, 5.0, and 5.4 [170]. The formation of polyelectrolyte complex nanoparticles is only possible if care is taken to ensure that the polymer solutions are well-dissolved, by filtering the solutions, as any undissolved polymer particles may nucleate the precipitation of the nanoparticles.

The ionic strength of the solution can be an important parameter controlling the formation of polyelectrolyte complexes. Bead-like particles can be prepared by dropping alginate (1.25% *w*/*v*) into chitosan (1.0% *w*/*v*, pH 1.0) containing 10% *w*/*v* NaCl. The sodium chloride supports the formation of spherical chitosan/alginate beads. Uncontrolled precipitation occurs when alginate aliquots are dropped into an aqueous chitosan solution without sodium chloride [87]. High ionic strength provides electrostatic screening, which helps to prevent the formation of large and uncontrolled precipitates. Chitosan microparticles can easily be prepared by dropping aqueous chitosan solutions into alkaline solutions. In this case, particles are created by precipitation, as chitosan is insoluble at a high pH [163].

Porous hydrogels are created using more concentrated polyelectrolytes solutions, followed by materials decantation, washing, and often the freeze-drying process. Porous chitosan/hyaluronic acid gels were obtained in the presence of sodium and calcium ions. Higher calcium concentration decreases the average pore size because calcium ions shield the carboxylate anions in the hyaluronic acid network [171]. Porous chitosan/alginate gels were also prepared by controlling the ionic strength, using different sodium chloride contents (0, 0.15, and 0.5 M) at pH 4.0 to guarantee ionized polysaccharides in aqueous solutions. The ionic strength significantly influences the Zeta potential due to electrostatic screening [172]. Porous chitosan/chondroitin sulfate coacervates (hydrogels) were prepared by dropping an aqueous chondroitin sulfate solution (25% *w*/*v*) into 1.6% *w*/*v* chitosan solution created in 0.57 M aqueous hydrochloric acid solution. The chitosan/chondroitin sulfate weight ratio should be adjusted to support stable assemblies with high porosity [173].

The disadvantages of these complexes are often their weak mechanical properties and low durability. They are prepared mainly by the establishment of ionic forces between polyelectrolytes. This complexation often results in brittle materials with structural heterogeneity. Moreover, the ionic crosslinks supporting the structures can readily be reversed in aqueous media with high ionic strength. Because of these traits, it is challenging to control the material porosity and pore size. Physical assemblies in water and biological fluids reorganize due to polymer chain self-assembly. Durability can be enhanced by using polymers with high molecular masses, which introduce effective polymer entanglements. However, it is difficult to prepare chitosan solutions above 2.0 wt.%, even using chitosan of low molecular weight (<100 kDa). To overcome this challenge, concentrated chitosan (4.0% *w*/*v*) solutions can be prepared in ionic liquids. Chitosan/*iota*-carrageenan and chitosan/chondroitin sulfate complexes were prepared in the ionic liquid 1-hydrogen-3-methylimidazolium hydrogen sulfate. However, ionic liquids have many disadvantages for biomedical-engineering applications, including high cytotoxicity and complex syntheses [174,175]. They can be retained in the resulting complexes, even after performing washing steps by Soxhlet extraction.

### 3.3. Ionotropic Gelation

Marine polysaccharides (especially the anionic alginates and carrageenans) can be associated with metallic cations. Physical materials (e.g., films, hydrogels, beads, and nanoparticles) are created following the ionotropic gelation method. Multivalent metal ions interact with the ionized carboxylate and sulfate groups in the polysaccharides, stabilizing the materials. Chitosan can also be associated with sodium tripolyphosphate for providing nano- and microparticles by the ionotropic gelation method. The phosphate sites on tripolyphosphate stabilize the chitosan chains by forming electrostatic interactions with the protonated amino groups at a pH range between 3.0 and 6.0. These materials are prepared at mild pH and temperature conditions, and avoiding the use of toxic chemistries (crosslinkers, additives, and surfactants) [8,78,80,176,177,178]. These traits are essential for preparing materials for sensitive biomolecule delivery, including proteins, nucleic acids, and cells [80,84].

Microfluidic approaches have been demonstrated for the generation of size-controlled alginate-based nanomaterials by ionotropic gelation. Nanomaterials that encapsulate antibodies, cells, and proteins have been prepared following one-step strategies in situ [80,82,90,179]. Microfluidic approaches control the nanoprecipitation of alginate nanoparticles in the presence of divalent cations by adjusting (i) flow rate, (ii) polymer concentration to achieve a desirable material size with low polydispersity index, and (iii) mixture composition to control the drug loading efficiency and release behavior. By precise control of the particle size and mixing, materials prepared by microfluidics have demonstrated distinct advantages compared to other nano- and micro-sized alginate materials (Figure 4C) [88,179,180,181].

### 3.4. Solvent Evaporation, Cooling of Polymer Solutions, and Freezing–Thawing

The solvent evaporation of polymer mixtures (blends) creates films following the “casting” method. These can easily be prepared by pouring polymer solutions into Petri dishes, and allowing the solvent to evaporate. After solvent evaporation, free-standing polymer films can be obtained by peeling them off of the Petri dishes. The polymer concentration and solvent type alter the polymer solution rheological properties. These properties influence the solvent evaporation rate, changing the film thickness. The solvent evaporation method presents some advantages, including low-cost and simplicity of processing; this method results in homogeneous films with controlled thickness by tuning the polymer concentration. The disadvantages include the use of organic solvents and inorganic acids to obtain polymer blends, and they may result in brittle polysaccharide-based materials that require plasticizers to achieve suitable mechanical properties [182,183].

The thermal behavior of some polysaccharides in solution can be exploited to prepare thermo-responsive hydrogels, including films [184] and porous materials [145]. Some carrageenans and gums undergo a coil-to-double helix transition that can induce gelation [185]. The gelation of polysaccharide-based solutions (e.g., carrageenans and gellan gum/chitosan mixtures) occurs at low temperatures (between 4 and 25 °C). Cations, including metallic ions and chitosan, stabilize this conformation, supporting hydrogel formation after the polymer solution is cooled. This straightforward process is useful for engineering scaffolds and injectable materials for drug delivery purposes [186]. The disadvantage is that polymer solutions are often prepared at high temperatures, preventing encapsulation of thermally sensitive payloads, such as cells and GFs into the materials.

The freezing–thawing method provides physical materials through consecutive freezing-and-thawing cycles of polymer solutions [187]. This method can produce durable matrices without any chemical crosslinking agents [188]. During the freezing steps performed in temperatures lower than 0 °C, the solvent crystallizes, concentrating the polymer chains in regions between the solvent crystals, and promoting physical junction zones between the polymer networks [189]. Consecutive freezing–thawing fosters self-assembling of polymer chains, and stable assemblies are achieved mainly by H-bonding interactions [148]. This method often can produce porous materials (hydrogels) containing polysaccharides for biomedical applications [190].

### 3.5. Fibers by Electrospinning of Polymer Solutions

The electrospinning of polymer solutions can be used to form micro and nanofibers. The electrospinning process occurs when an electric field is applied between the tip of a capillary needle containing a polymer solution and a grounded metallic collector, under a pre-established flow rate. The electric field between the tip and the collector applies tension that should overcome the polymer solution surface tension, elongating the polymer solution drop to a conical shape, called a Taylor cone. When the electric field reaches a critical value, in which the repulsive electrical forces exceed the surface tension, the polymer solution is ejected from the tip of the Taylor cone to the grounded collector. The solvent is evaporated during the fiber trajectory to the metallic collector. The electric field controls the trajectory of the charged polymer jet, but instabilities caused by changes in the mechanical properties as the solvent evaporates and the polymer precipitates, cause the jet to “whip”. This whipping motion stretches and draws the jet into a very thin fiber, producing polymer fibers [191,192,193].

Electrospun fibers have a high surface area to volume ratio, high porosity, desirable mechanical properties, and surface functionality compared to conventional fibers, films, and hydrogels. Polysaccharides are blended with synthetic and semisynthetic polymers for producing electrospinnable solutions and electrospun materials with good mechanical properties, including scaffolds [194], wound dressing [195], and DDSs [196]. However, in some cases, the biological application of electrospun materials is not recommended because polymer solutions are often created in toxic organic solvents that may be retained in trace amounts in the fibers. This can require additional washing steps for biomedical applications.

### 3.6. Three-Dimensional Bioprinting

Three-dimensional (3D) bioprinting creates structures that mimic the organization and structures of extracellular matrices of organs for tissue restoration. Moreover, 3D bioprinting enables the deposition of cells and GFs directly during bioprinting to precisely tune or pattern the locations of specific moieties, mimicking their organization in vivo. The main objective is to create scaffolds that mimic native tissues and organs; in principle this can include printing complex structures containing multiple different compositions and cell types to recapitulate organ structures [197]. Some factors can influence material printability, including polymer concentration, printing pressure, and printing speed [198]. These factors play a critical role in the biological and mechanical properties of the 3D bioprinted materials.

Polysaccharide-based bioinks have been used to create 3D bioprinted scaffolds. Bioinks are often hydrogels containing live cells in a suitable cell culture medium [199,200]. Hydrogels are the principal materials applied as bioinks for the regeneration of neural [201], cardiac [202], cartilage [203], and skin [204] tissues.

## 4. Biomedical Applications

This section reviews polysaccharide-based assemblies applied as scaffolds, wound dressings, surface coatings, and DDSs for GF delivery [54,205,206]. Cytocompatible scaffolds can be formulated into 3D porous structures with controlled biodegradation rates in biological environments. Scaffolds and wound dressings should prevent microbial adhesion and growth to prevent infection and to promote normal healing processes. Coatings are engineered to cover biomedical devices implanted in the body, and are designed with surface chemistries that mitigate undesirable responses, such as inflammation and the foreign body reaction, while promoting the healthy integration with surrounding host tissue. These materials should also prevent biofilm formation, via antimicrobial and antiadhesive properties. These properties should be application-specific. For example, for wound-healing applications, coatings and scaffolds should support hemostasis, or the formation of a blood clot. However, for many cardiovascular applications, such as blood-vessel engineering, heart valves, and arterial stents, surfaces should be designed to prevent blood clotting. Hemocompatible materials are designed to reduce protein adsorption, prevent platelet adhesion and activation, reduce hemolysis, and prevent inflammation. DDSs for GFs should protect the GFs and present the GF in a context that mimics its biological presentation through sustained release and through other biochemical signals that promote growth factor signal transduction. Materials with these properties are projected for biomedical applications [207].

### 4.1. Physical Assemblies as Scaffolds without GFs

Polysaccharide-based scaffolds are porous materials with 3D structures. Polysaccharides are excellent candidates for biomedical applications because they mimic extracellular matrix composition and functions, provide cytocompatible and supporting microenvironments for tissue healing and repair [208,209]. The extracellular matrix (ECM) is composed of fibrillar proteins (collagen, elastin, fibronectin, and laminin), glycosaminoglycans, and proteoglycans (proteins with covalently bonded glycosaminoglycan side chains). It mediates metabolite transport due to an interconnected pore network and provides mechanical support for cell orientation, migration, growth, and differentiation [208]. Therefore, the ECM modulates the so-called cell-fate decisions, whereby progenitor cells terminally differentiate into mature tissue cells.

Table 1 presents the principal approaches used to create physically assembled scaffolds used to support repair of bone, skin, cartilage, epithelial, and neural tissues. Polysaccharides contain hydrophilic groups (-OH, -NH_2_, -OSO_3_H, and -COOH) that bind proteins and cells to promote anchorage. Similarly, the fibrillar proteins and glycosaminoglycans in the ECM are either negatively charged (glycosaminoglycans, collagen) or positively charged (elastin, which has a high pK_a_). Many mammalian cell types attach and grow well on negatively charged surfaces, such as glass and tissue culture polystyrene; surfaces with protonated amine moieties can also support the attachment and growth of mammalian cells and are superior to negatively charged materials for some cell types. Protonated surfaces may electrostatically interact with the negatively charged ECM, which contains sulfated glycosaminoglycans [195,210]. Interactions between charged moieties in the ECM and cationic groups on chitosan-based assemblies may improve adhesion, proliferation, and spreading of some cells [211].

Figure 9 shows chitosan-based scaffolds (porous scaffolds, films, and electrospun fibers) applied in mammalian cell culture. Chitosan is the unique natural polysaccharide with cationic behavior in aqueous solutions at pH lower than 6.5 [208]. Chitosan/gellan gum scaffolds created by gelation of polymer blends provide porous structures after the freeze-drying process [145]. X-ray photoelectron spectroscopy indicated that protonated amino groups occur on the scaffold surface even after the washing step. Scaffolds prepared with the highest chitosan content (80 wt.%) promote anchorage, growth and spreading of bone-marrow-derived mesenchymal stem cells after 9 days of exposure (Figure 9). By controlling the chitosan content in the blends, the pore size (148 µm) and porosity can be tuned. The wet physical assembly composed of 80 wt.% chitosan has a Young’s modulus of 470 Pa. Higher gellan gum content decreases the scaffolding capacity toward mammalian cells. The cells do not spread and proliferate on physical materials at a 60/40 chitosan/gellan gum weight ratio. By increasing the chitosan concentration from 60 to 80 wt.%, the water uptake capacity also reduced from 44,960% to 2603%. A high water uptake seems to prevent the attachment of cells to the physical assembly. Interconnected pore networks can support vascularization, migration, and proliferation of bone cells; however, the material composition needs to be adjusted to optimize scaffolding capacity [145].

Porous chitosan/chondroitin sulfate and chitosan/alginate assemblies can also be optimized to support mammalian cells for tissue scaffolds. Suitable mechanical properties, porosity, and structural homogeneity were achieved at 4.0 *w*/*v*. chitosan and 1.0 *w*/*v*. chondroitin sulfate or 1.0 *w*/*v*. alginate [208,213]. Chitosan/hyaluronic acid scaffolds had higher mechanical properties at 8.0 wt.% hyaluronic acid than the pure hyaluronic acid-based scaffold [216]. Another binary physical assembly composed of chitosan (40 wt.%) and ulvan (60 wt.%) mediated the formation of globular structures of apatitic minerals, demonstrating that they can support mineralization required for bone healing. The scaffold promoted cell attachment, proliferation, and osteogenic differentiation of pre-osteoblast cells, inducing ECM formation, which was suggested by the alkaline phosphatase activity and collagen production. The durable assembly also supported calcium phosphate mineralization [219].

Scaffolds based on chitosan/pectin films prepared from the solvent evaporation method also fostered the anchorage, growth, and spreading of human adipose-derived mesenchymal stem cells after 7 days of exposure (Figure 9) [138]. Pectin with a high *O*-methoxylation degree (56%) produced durable films even in high pectin contents (higher than 70 wt.%) in the film. However, the scaffolding capacity was also achieved by controlling the chitosan/pectin (66/34 *w*/*w*) weight ratio in the polymer blend before the solvent evaporation. The water uptake capacity played an essential role in the biological responses. The chitosan content regulates the surface wettability, providing suitable conditions to attach cells. The 66/34 chitosan/pectin weight ratio produced hydrophilic films (water contact angle of 61.7° after 15 min of a water droplet contact with the film surface), high surface roughness, and lower water uptake compared to the other films created with a pectin content higher than 66 wt.%. Moreover, this condition produced a film with an ultimate tensile strength of 28 MPa [138]. This result is similar to the tensile strength of human skin (between 5.0 and 30 MPa, depending upon the orientation and location) [231]. Therefore, a film that mimics the mechanical properties of skin was created, and its properties can be tuned by modulating the chitosan content in chitosan/pectin polymer blends.

Polysaccharide-based materials often have weak mechanical properties, and chitosan-based assemblies need to be washed to remove residual H_3_O^+^ contents [138,145]. Overall, the mechanical properties of the physical assemblies are improved by associating them with gelatin [54,195], γ-polyglutamic acid [225], polyethylene oxide [195], poly(ε-caprolactone) [206,223,224], poly(ethylene glycol) [229], poly(vinyl alcohol) [227,228], polyethyleneimine [226], and Manuka honey [205]. For example, Manuka honey (2.0 *w*/*v*) significantly increases the elastic modulus of gellan gum scaffolds stabilized with Ca(II) ions to 116 kPa. The cytocompatibility is achieved by controlling the content of gellan gum and Manuka honey in the polysaccharide-based assemblies.

Synthetic materials do not generally contain the rich variety of biochemical signals present in the natural ECM. In some cases, adding synthetic materials can reduce the scaffolding capacity of polysaccharide-based assemblies compared to the materials composed of natural macromolecules only. Scaffolds based on polyethyleneimine and plasmid DNA showed that the Schwann cell viability reduced as the concentration of polycation polyethyleneimine in the scaffold increased [232]. Other polycations, including cationic tannin derivative (an amino-functionalized polyphenol tannin material called Tanfloc, pK_a_ = 6.0) [221], gelatin (especially type A with an isoelectric point around 7.0) [195], and ε-polylysine (a homo-polyamide, pK_a_ = 9.3–9.5) [222], can replace chitosan to provide physical assemblies [226]. Mechanical properties are essential for supporting mammalian cells and can provide differentiation signals through mechanotransduction. However, other features, including surface wettability and roughness, pore structure, and hydrophilic–hydrophobic balance, should also be tuned to optimize biological responses to scaffolds [138,145,195].

### 4.2. Surface Coatings and Thin Films without GFs

Polysaccharide-based coatings can be prepared via the assembly of polyelectrolyte multilayers (PEMs) on solid substrates (Figure 10A). PEMs have been used to prevent microbial infections and often support cell adhesion, proliferation, and differentiation on solid substrates engineered for biomedical applications. PEMs with antimicrobial activities are usually composed of chitosan and chitosan derivatives containing protonated ammonium sites (*N*-quaternized moieties). These groups interact with negatively charged cell walls of Gram-positive and Gram-negative bacteria, which are mainly composed of the anionic dipalmitoyl phosphatidylglycerol and other negatively charged phospholipids. Cationic polymers increase the membrane permeability, leading to the leakage of intracellular materials (glucose, nucleic acid, and lactate dehydrogenase), preventing the transport of nutrients to microbial cells, causing cell death [37,233].

Chitosan can form PEMs assembled with hyaluronic acid [234], heparin [235], *iota*-carrageenan, and pectin [37] as polyanions, resulting in antimicrobial and anti-adhesive properties toward both Gram-positive and Gram-negative bacteria. Chitosan and heparin impart bactericidal activity, while heparin and *iota*-carrageenan confer the anti-adhesive behavior [236]. For example, Martins and coworkers developed polyelectrolyte multilayers (15 layers) of *iota*-carrageenan/chitosan and pectin/chitosan by layer-by-layer deposition on oxidized glass substrates. The materials exhibited excellent anti-adhesive and bactericidal activities against *Pseudomonas aeruginosa* (*P. aeruginosa*, Gram-negative) and *Staphylococcus aureus* (*S. aureus*, Gram-positive) [37]. Figure 10B shows SEM images of the PEMs seeded with the *P. aeruginosa* after 6 h of exposure. Compared to the control samples (a native polystyrene film for cell culture), the chitosan/*iota*-carrageenan PEM (15 layers) significantly prevents the attachment of microbial cells, killing the adhered cells after 6 h (Figure 10B). Polysaccharide-based PEM coatings promoted a considerable reduction of bacterial adhesion compared with the polystyrene (control) [37]. The higher wettability and the negative charge density supported by -OSO_3_^−^ (on *iota*-carrageenan) and -COO^−^ (on pectin) groups on PEMs contributed the anti-adhesive property. The anti-adhesive property is a feature of hydrophilic PEMs enabling them to avoid microbial attachment and growth [37]. Chitosan/heparin PEMs also provided anti-adhesive and antibacterial traits to amino-modified poly(ethylene terephthalate) films. The bactericidal action depended on the pH condition in which the PEMs were assembled [235]. A significant *E. coli* inhibition was observed for PEMs assembled at pH 3.8 compared to PEMs assembled at higher pH (between 4.0 and 6.0). Free H_3_O^+^ ions can kill microbial cells, as well. Moreover, low pH supports more protonated amino sites in the chitosan, enhancing its bactericidal property [237].

Follmann et al. evaluated the anti-adhesive and antibacterial properties of *N*,*N*,*N*-trimethyl chitosan/heparin PEMs on oxidized polystyrene substrates [233]. The antimicrobial activities mainly depended on the *N*,*N*,*N*-trimethyl chitosan quaternization degree. *N*,*N*,*N*-Trimethyl chitosan 80% quaternized had the highest biocide action against *E. coli* due to the high content of ammonium groups on the PEM [233].

An amino-functionalized polyphenolic tannin derivative (called Tanfloc) has excellent properties to be used in biomedical applications. It can replace chitosan to provide antimicrobial PEMs. Facchi et al. assembled a polyphenolic tannin derivative (Tanfloc) with pectin and *iota*-carrageenan at pH 5.0 on oxidized glass. The PEMs significantly prevented the attachment and proliferation of *S. aureus* and *P. aeruginosa* after 24 h of exposure [239]. Compared to chitosan/heparin PEMs (10 layers), the Tanfloc/heparin PEMs (10 layers) deposited on oxidized glass support hemocompatible surfaces. The surface has antifouling properties by preventing blood serum protein (fibrinogen) adsorption and platelet adhesion and activation [154]. Figure 10C shows that the Tanfloc/heparin PEM (10 layers) significantly avoids platelet adhesion compared to the chitosan/heparin PEM. The calcein-AM stained the adhered platelets (green) on the PEM surface. The authors suggested that polyphenolic moieties on Tanfloc support a pseudo-zwitterionic effect and catechol moieties may both provide anti-platelet adhesion feature [154]. Therefore, amino-functionalized tannin derivative and the anti-adhesive and anticoagulant heparin activities impart the antimicrobial [239] and hemocompatible properties to the Tanfloc/heparin PEMs [154]. Hemocompatible PEMs can be deposited on biomedical devices (catheters, stents, and others) to prevent blood clotting in biomedical implants.

The Tanfloc/heparin PEMs (five layers) deposited on titania nanotubes also demonstrated scaffolding capacity toward adipose-derived mesenchymal stem cells after 7 days of exposure. The PEMs and the native titania nanotube surfaces promote the adhesion and proliferation of mammalian cells. Figure 10D presents fluorescence images of the surfaces seeded with the adipose-derived mesenchymal stem cells after 7 days. The cells were stained with DAPI (blue) and rhodamine-phalloidin (red). Both surfaces have scaffolding capacity (Figure 10D). However, the Tanfloc/heparin PEMs induced the adipose-derived mesenchymal stem cell differentiation after 3 weeks. Compared to the native titania nanotube surfaces, the PEM provided higher osteocalcin deposition than the unmodified titania nanotube (Figure 10E). The percentage area coverage of osteocalcin is higher (80%) than the coverage area on the titania nanotubes after 3 weeks [140]. The Tanfloc/heparin PEMs have enhanced osteoinductivity toward the adipose-derived mesenchymal stem cells.

### 4.3. Growth Factor Delivery for Tissue Repair

GFs are recognized as important natural signaling agents that guide wound healing and tissue morphogenesis. GFs are powerful signaling proteins that affect cell migration, cell proliferation, stem cell differentiation, and ECM production, leading to tissue healing and morphogenesis. They comprise many families of proteins and are secreted by multiple cell types during wound healing. GFs act on target cells through both autocrine and paracrine mechanisms binding to cell surface receptors or growth factor receptors (GFRs). Their actions are timed and orchestrated so as to initiate, coordinate, and resolve various stages of wound healing, and to ensure that newly formed tissues are organized and functional. GFs have therefore been proposed as important biochemical signals to include in tissue engineering and regenerative medicine strategies. The activity of these potent proteins is tightly regulated by the kinetics of their release, the biochemical context of their presentation, the expression of their cognate GFRs, and their relative instability. These features of GF signaling enable GFs to potently act very locally in space and time during normal tissue morphogenesis and wound healing.

Some tissue healing processes are regulated by a cascade of signals involving multiple GFs acting on multiple cell types. GFs may recruit cells to a cite of injury, drive their differentiation, and promote cell organization and extracellular membrane deposition. Some GFs that act as chemoattractants and mitogens, and are involved in healing and regeneration of many tissues. Other GFs drive the differentiation of only specific cell types, or maintain the function of certain differentiated cells. While the goal of GF delivery in tissue engineering and regenerative medicine is typically to promote anabolic (or anticatabolic) processes, some GFs also drive necessary catabolic processes, enabling cell migration, for example, or creating space for the deposition of new extracellular membrane.

Table 2 lists tissues and healing process mentioned in this review, and some of the important GFs known to affect them. For example, angiogenesis, or the formation of new blood vessels involves vascular endothelial GF (VEGF), placental GF (PlGF), insulin-like GF (IGF), fibroblast GFs-1 and -2 (FGF-1 and FGF-2), transforming GF (TGF), platelet-derived GF (PDGF), and hepatocyte GF (HGF) [240]. Bone-fracture healing also involves multiple GF signals occurring in sequence, including members of the FGF family, multiple PDGF isoforms, and several members of the TGF-β superfamily [241,242,243].

While GFs hold great promise as therapeutics to treat diseased and injured tissue, fine control over their delivery may be required to take full advantage of their therapeutic potential. GF delivered in high doses, with sustained activity, or in the wrong location may have undesirable side effects. For example, the overexpression of several GFs is a hallmark of many tumors [253,254]. The GF bone morphogenetic protein-2 (BMP-2) induces ectopic-bone formation when delivered subcutaneously [255], and transforming growth factor-beta (TGF-β) signaling is involved in fibrosis associated with multiple pathologies and diseases [256]. Therefore, GF delivery strategies must ensure that GFs act in the right place at the right time, and it may be just as important to ensure that the GFs do not act in the wrong place at the wrong time. Controlled GF delivery must also ensure that the correct amount of GF is presented. GFs typically have non-monotonic dose responses that have an optimal range, or biphasic dose responses that cause different cell responses depending upon the GF concentration [257,258]. For example, VEGF concentrations below or above an ideal dose can result in the formation of leaky blood vessels [259]. Therefore, when designing GF delivery vehicles care must be taken to control the dose, timing, and location of the delivered GF.

Many GFs are known to be particularly unstable. Therefore, materials (hydrogels, coatings, films, nanoparticles, and others) for controlled GF delivery should first be capable of stabilizing their precious payload. As an example, the TGF-β superfamily of GFs and the neurotrophins are active only in their dimeric forms [244,249]. These can degrade by multiple mechanisms, including aggregation and disulfide bond rearrangement [260]. FGF-2 and VEGF are both rapidly degraded by serine proteases found in blood [261,262]. FGF-2 and NGF for example, have reported half-lives of less than 5 min after intravenous injection [263]. Some FGFs have notoriously low thermal stability, and begin denaturing, at temperatures below body temperature [264,265]. FGF-1 is about 30% unfolded at 37 °C [266]. Thermal instability can lead to rapid loss of GF activity at normal cell culture conditions [264]. The instability of GFs in vivo helps modulate their activity, enabling tight control over their functions. However, this instability presents a significant challenge for formulating controlled GF delivery vehicles that can achieve signaling over the time scales necessary for tissue healing.

Their relative instability makes single-dose administration of GFs ineffective. The administration of GFs can provide a high concentration in the body in an initial burst stage, extending above the maximum desired level in the pharmacokinetic curve; however, after a short time, the GFs concentration rapidly decreases in the body (Figure 11). Therefore, GFs should be formulated with DDSs, including scaffolds, surface coatings, nanoparticles, and microparticles (Figure 11). These materials should be designed to provide stability to the GFs for tissue-engineering applications, by protecting them from degradation and sustaining their release to targeted cells or tissues (Figure 11) [257,267]. The GF’s bioactivity can also be extended by the synergistically binding GFs and GF receptors, thereby promoting GF signaling pathways. This effect is achieved by the interaction between the biomaterial/GF pair and specific GF receptors on cell surfaces. As a result, the GF efficacy can be enhanced [257,267].

Materials for controlled GF delivery should be able to present the GF to their receptors in a way that effectively actuates the downstream signaling mechanisms [269]. This can be achieved by consideration of the biological context in which GF signaling occurs. GFs are produced and sequestered in the ECM. ECM binding of GFs helps to maintain their stability and provides a reservoir of GF that can be accessed during wound healing. The ECM can thereby act as a ready store for the on-demand presentation of a GFs in response to tissue injury. The components of the ECM also help to organize these interactions at multiple length scales. At the subcellular scale of macromolecules and macromolecular complexes, the ECM can facilitate protein–protein interactions. In the case of GFs, components of the ECM can stabilize or inhibit GF binding to the cognate GF receptor (GFR). The ECM may also promote GF or GFR oligomerization, which can amplify GF signaling. Through these mechanisms, the natural ECM is a modulator of GF activity at the subcellular level. The stabilization of GF–GFR binding can also be mimicked by rationally designed materials for GF delivery. Materials for GF delivery can also facilitate GF activity by mimicking other functions of the ECM at larger length scales. For example, at the cellular level, the ECM guides cell polarization and migration, and coordinates homotypic and heterotypic cell–cell junctions. At the tissue level, the ECM provides structure that organizes cell types into functional domains. GF delivery from three-dimensional tissue scaffolds may be ineffective, if the scaffold does not support or promote these interactions at the cellular and multicellular length scales. Design of materials for GF delivery should include a rational approach to selecting which, if any of these important features of the ECM to mimic.

The desired mode of action of a GF may dictate how it should be presented or delivered. The slow release of a GF can be used to create gradients (by diffusion or diffusion combined with degradation), to recruit cells to a site of wound healing. On the other hand, for some applications, the stable presentation of a surface-bound GF may be more effective for promoting some GF activities [257,267]. The delivery of multiple GFs together or in sequence may further improve control over GF activity.

In summary, GF delivery vehicles must be capable of stabilizing relatively unstable GF molecules from chemical, enzymatic, and thermal degradation, over time scales appropriate to the desired signal. GF delivery strategies must deliver the right amount of GF to the right place at the right time, while avoiding off-target activities and side effects due to improper dosing. This may involve delivering multiple GFs with different release profiles, or developing stimuli-responsive materials that can present a GF on-demand, in response to a biological cue. Some GF delivery objectives may require the controlled release of a GF, while other applications might be more effective when the GF is stabilized and presented at a surface or interface. Effective GF delivery may be facilitated by materials that can also bind cognate GFRs, and that enable cells to organize across multiple length scales. In the next section, we will discuss how physical polysaccharide-based materials are suitable for meeting these demanding challenges of GF delivery.

#### Polysaccharides for Controlled GF Delivery

As reviewed above, some polysaccharides have excellent processing characteristics. They can be formulated into films and coatings, porous foams, fibers, hydrogels, and tablets. Polysaccharides can be readily assembled into nanomaterials [270], including ultrathin coatings, nanoparticles, hydrogels, and fibers. These structures can mimic the macromolecular organization and dimensions of the features of the ECM in which native polysaccharides interact with GFs and cells. Natural polysaccharides at surfaces and in solution also exhibit inherent antimicrobial, antimycotic, and bactericidal activities. Finally, some polysaccharides provide specific and nonspecific sites for binding other ECM components, including structural proteins and the mineral phase of bone tissue. They also provide both direct and indirect (e.g., mediated through other bound proteins) sites for cell adhesion or even selective cell adhesion. Cell adhesion is a prerequisite for cell migration and proliferation.

Polysaccharides have inherent GF-binding sites that can stabilize GFs and GF-GFR complexes. Polysaccharides may also demonstrate non-specific GF binding, and protection of GFs from multiple modes of degradation. As scaffold materials they can be readily formulated into a variety of extracellular membrane–mimetic structures with tunable biodegradation; they present adhesion sites for mammalian cells, inhibit bacteria and other pathogens, and organize other components of the extracellular matrix. The combination of these properties in a single class of materials far exceeds the combination of functions that could be designed into synthetic multifunctional materials. Since polysaccharides are important functional features of the extracellular and pericellular space, where GFs directly interact with cells, polysaccharides can present GFs in a context that mimics many of the biological aspects of the natural presentation of GFs.

In 1983 the polysaccharide heparin was recognized as a co-factor, enhancing the mitogenic activity of a growth factor [271]. Upon finding that the GF was a heparin-binding protein, Maciag et al. hypothesized that heparin binding may stabilize GF tertiary structure, or even reactivate inactive GF [272]. This GF had been known as endothelial cell GF, and was later identified as FGF-1 [273]. Since this early work, the list of heparin-binding GFs has grown substantially. Ori et al. curated a list of 435 individual heparin-binding (and heparan sulfate-binding) human proteins [274]. This list includes many of the GFs of interest in tissue engineering, including several BMPs (BMP-2, -3, -4, -6, and -7); 17 out of the 22 known human FGFs; both VEGF-A and –B; both PDGF subunits (PDGF-A and PDGF-B); TGF-β1 and -β2; and hepatocyte growth factor (HGF). Other heparin-binding GFs on the list include connective tissue GF (CTGF); leukocyte-derived GF (LDGF); hepatoma-derived GF (HDGF); placental GF (PlGF); granulocyte-macrophage colony-stimulating factor (GM-CSF); and interleukins IL-2, -3, -4, -5, -6, and -7. Interestingly, this list of heparin binding proteins also includes other proteins important for GF activity, such as all four FGF receptors (FGFR-1, -2, -3, and -4), HGF receptor, VEGF 165 receptor, and VEGF receptors (VEGFR-1 and -2), among others. VEGF delivery is controlled naturally by it sequestration by other extracellular membrane molecules [259]. This vast repertoire of GF and GFR binding of heparin enables heparin to mediate many GF signals.

Perhaps the most well characterized GF-heparin interactions are those with FGF-2. The binding of FGF-2 to heparin occurs through specific sulfation patterns on the heparin binding to a known site on FGF-2 [275]. The binding of FGF-2 to heparin protects the GF from loss of activity in cold storage, at high temperature, and from acid catalyzed degradation [276]. Furthermore, the heparin stabilizes the FGF-2-FGFR-1 complex, promoting receptor dimerization and thereby facilitating GF signal transduction [277]. The protection afforded by binding to heparin extends to other sulfated polysaccharides as well, including sulfonated dextran, dextran sulfate, *lambda*-carrageenan, and chondroitin sulfate [262,278]. Heparin has similar effects on other growth factors. For example, heparin has been shown to bind and protect TGF-β1 [279], and to enhance the binding of VEGF to its cell surface receptors [280].

Almodovar et al. showed that PEMs of heparin and chitosan could be used to deliver FGF-2 to mammalian cells from surfaces such as glass and medical-grade titanium. When the FGF-2 was presented to the cells bound to heparin in the PEM, its ability to promote cell proliferation was enhanced compared to delivery of the same growth factor at an optimal dose in solution [257]. Place et al. built on this work to prepare materials that mimic the structure and composition of proteoglycans in the ECM. Proteoglycans in the ECM bind and stabilize the FGF-2 and also facilitate GF signaling by forming a ternary complex with the GF and its cell surface receptor. Place et al. prepared graft copolymers composed of a modified hyaluronic acid backbone, with covalently attached glycosaminoglycan side chains, and used them to deliver FGF-2, demonstrating enhanced FGF-2 activity compared to delivery of the growth factor in solution [269]. Place et al. also prepared polyelectrolyte complex nanoparticles using heparin or CS as the polyanion and chitosan or trimethyl chitosan as the polyanion. Proteoglycan mimics prepared using chondroitin sulfate and either chitosan or trimethyl chitosan could stabilize FGF-2 for up to two weeks in cell culture media, and this stabilization was similar to what was achieved by binding the growth factor to the PG aggregan (which is rich in chondroitin sulfate) [75]. Similar PG mimics prepared using heparin instead of CS performed even better than aggregan with respect to FGF-2 stabilization and delivery.

The incorporation of GF directly into scaffolds for tissue engineering can be challenging, because the GF requires special handling and mild processing conditions to retain its stability. To incorporate FGF-2 into electrospun nanofibers, Place et al. investigated several techniques, including electrospinning emulsions and coaxial electrospinning of two different solutions by using a compound needle. They found that by binding FGF-2 to heparin in PG-mimetic polyelectrolyte complex nanoparticles, the FGF-2 could be incorporated into the electrospun nanofibers, and its biological activity could be preserved [281]. Zomer-Volpato et al. used a different approach. They adsorbed similar PG-mimetic nanoparticles to chitosan electrospun nanofiber networks after electrospinning, to deliver FGF-2, and demonstrated that FGF-2 activity could be at least partially preserved for over four weeks [267]. The preservation and maintenance of GF biological activity is important for developing tissue-engineering applications, as the time scales of wound healing in many tissues are measured in weeks or months, whereas the half-lives of growth factor stability in vivo may be measured minutes or hours [263].

Romero et al. used heparin-containing nanofibers to deliver TGF-β and FGF-2 from the surfaces of bone allografts in a mouse femoral defect model. While the GF delivery did not result in improved overall bone healing, it did reduce inflammatory responses [282]. Lin et al. also demonstrated that heparin-based PEMs could be used to deliver TGFβ to primary liver cell cultures in vitro as a technique for enhancing maintenance of the primary hepatocyte culture [283].

While heparin plays many known roles in GF signaling, other glycosaminoglycans are also important. Chondroitin sulfate and dermatan sulfate are abundant polysaccharides in the ECM that influence GF activity. Heparin binding GFs have also been shown to bind to chondroitin sulfate E [284]. Chondroitin sulfate has been particularly associated with brain development, where it regulates the activities of FGF-2, HGF, and pleiotrophin [285]. Heparan sulfate-based proteoglycans are principal compounds in the ECM. They control cellular functions and some biological responses, being co-receptors for anionic fibroblast GFs. Heparan sulfate proteoglycans have a higher affinity (6-fold) toward fibroblast GFs than pure heparin [24]. The receptor binding can be modulated in drug delivery systems to promote slower and sustained GF delivery. Proteoglycans with highly sulfated chondroitin sulfate (types D and E) repair neural tissues, while the other GAGs have no similar capacity. Therefore, the sulfated GAGs with higher sulfation degrees have demonstrated better biomedical properties. This suggest that GAGs with a high sulfation degree could be important for supporting tissue healing in tissue-engineering applications [31].

While heparin and other glycosaminoglycans can bind GFs through specific ionic interactions, polysaccharides in general can also stabilize GFs by other mechanisms. GFs may bind to polysaccharides through non-specific ionic and H-bond interactions. Heparin and brain natriuretic peptide GFs mainly interact via H-bonds [286]. Binding to polysaccharides can promote GF stabilization. Small-molecule saccharides (e.g., trehalose, mannitol, and sorbitol) are well-known excipients used in food and pharmaceuticals. Their exact modes of action are not completely understood, but they may contribute to glass formation, thus preventing protein unfolding and increasing thermal stability. They may also provide H-bonds at the protein surface that help prevent chemical degradation and aggregation of encapsulated proteins. These exact modes of action of mono- and oligosaccharides may or may not extend to polysaccharides.

Nonetheless, as we review below, cellulose, chitosan, hyaluronan, alginate, and sulfated polysaccharides (chemically modified) have been shown to improve GF stability in a variety of contexts. The binding capacity depends on the sulfation degree, sulfate group location in the sugar units, molecular mass, and polymer conformation [104,108,109,287,288]. Sulfated marine polysaccharides provide attractive alternatives to prepare GFs delivery devices [97,105,289,290,291,292]. However, few works report the use of sulfated marine polysaccharides in GF delivery strategies.

*Kappa*-carrageenan can be used to prepare durable materials with gelling behavior in physiological conditions and biological fluids, suitable for forming injectable materials for cartilage [103,291,293] and bone repair [294]. Injectable *κ*-carrageenan-based materials release vascular endothelial GFs (VEGF) and platelet-derived GF (PDGF-BB), promoting healing and regeneration of tissues [294,295]. Sun et al. showed that *λ*-carrageenan binds basic and acidic fibroblast GFs (FGF2 and FGF1), protecting them against denaturation at ambient temperature during long-time storage [278]. Fucoidan binds GFs, controlling their release rate [296,297,298]. It has been used to prepare injectable adhesive materials loaded with GFs from platelet-rich plasma for cartilage repair [108] and diabetic foot ulcers [81]. These properties rely on fucoidan molecular mass as well. High molecular weight fucoidan interacts better with vascular endothelial GF (VEGF), reducing burst release and promoting biological cues for angiogenesis [299]. Fucoidan stabilizes GFs over 16 days [194]. However, there are no reports using ulvan in GF delivery strategies. The marine polysaccharide structures mimic extracellular membrane-based GAGs and proteoglycans. Therefore, we believe that biomaterials based on marine polymers should be further explored to engineer materials for regenerative medicine, mainly due to their affinity toward GFs [97].

Sodium and potassium ions in the body fluids or other media can exchange the metallic ions in the alginate-based hydrogels, leading to the material disintegration owing to the calcium ion release [83,176,300]. The ionic exchange increases the material solubility, favoring the release process of loaded drugs [176]. For drug delivery purposes, drug release rates can be controlled by modulating the degradation kinetics of alginates by controlling their molar masses and G/M ratios or performing oxidation of polymer chains [301]. However, controlling the burst-release and producing alginate-based materials with desirable mechanical properties remain as outstanding challenges [90,91,92,301]. The relative release rate of GFs and mechanical properties can be modulated by controlling the amount of alginate within materials and alginate molecular mass [90,301,302]. Alginates are also chemically modified or physically associated with peptides [303,304,305], proteins (collagen and gelatin [303,304,306], hyaluronic acid [307], and chitosan [59,308,309] to improve their biological responses and sustain activity of sensitive and therapeutic proteins in target sites [176].

Bacterial cellulose can be used to develop scaffolds and membranes for human fibroblast GF-2 (FGF2) delivery [112,118,131,310]. Microbial cellulose membranes quickly incorporated GFs in their structures, supporting controlled release over 10 days [118]. Microbial cellulose-based materials also provide a prolonged release of VEGF for two weeks [311].

Cellulose nanocrystals provided from a microbial source is advantageous for tissue engineering purposes [131]. Cellulose nanocrystals provide locally and sustained release of vascular endothelial GFs (VEGF) [312]. Moreover, when integrated into drug delivery materials, cellulose nanocrystals support mechanical reinforcement and both PDGF and VEGF release. Cellulose nanocrystals improve the mechanical durability of composite materials against degradation (hydrolytic and enzymatic), supporting controlled protein delivery [47,313].

Sulfated hyaluronic acid has increased binding strength to epidermal GFs compared to unmodified hyaluronic acid [314]. Furthermore, sulfated alginates interact better with cationic proteins than the unmodified alginates with weak carboxylate binders. This improved binding behavior can reduce the burst protein release, improving the encapsulation efficiency of proteins in drug delivery vehicles, and enhance the protein stability [136,315,316,317]. A water-soluble 2-*N*, 6-*O*-sulphated chitosan binds GFs, controlling the release rate of epidermal (HGF) and bone morphogenetic protein (BMP-2) [10,318,319].

Controlled protein delivery is often achieved through the use of biodegradable polymers. Tuning the rate of degradation is one means of controlling drug release rate. Many polysaccharides are biodegradable by enzymes in mammalian tissues. Their degradation products are generally non-toxic saccharides, which are expected to have minimal burden on metabolic processes.

Some polysaccharides also have known antioxidant activity, including chitosans, carrageenans, fucoidan, alginates, and many others [320,321]. Scavenging reactive oxygen species and reactive nitrogen species can promote wound healing and protect GFs from degradation.

### 4.4. Wound Dressings

Skin wounds are caused by chemical, physical, irradiation, or thermal sources [322]. Non-chronic wounds can take as long as 8 to 12 weeks to achieve complete skin healing. Skin damage starts to be recovered immediately after the injury, with five stages, including homeostasis, inflammation, migration, proliferation, and maturation [322]. Chronic skin wounds require more time to heal, and the recovery time depends on the environment, social setting, location of the of injury, and patient health [322,323]. Between 1% and 2% of the population have chronic wounds at least once during their life [324]. Chronic wounds require much more time (more than five months, and sometimes more than a year) to heal due to prolonged inflammation [234].

The primary function of a wound dressing is to protect the wound against dehydration and pathogen [323]. Additionally, wound dressings should not interact with the damaged tissue, providing desirable healing conditions without pain. Wound dressings should be non-toxic and non-allergenic, should protect the wound against pathogens and should provide absorption of wound exudates. They should also act as a dermal substitute, and be permeable to provide gas exchange for promoting normal tissue repair [324]. Wound dressings should be durable and flexible to prevent the need for restricting motion during healing, maintaining integrity during application [324,325]. The wound-healing process can be promoted by releasing bioactive molecules from the wound dressing that maintain favorable healing environments supportive of tissue regeneration. For example, Long and coworkers developed wound dressing constituted of 3D bioprinted chitosan-pectin hydrogel. They demonstrated lidocaine hydrochloride delivery for 4 h [326].

Polysaccharide-based materials have been extensively proposed as wound dressings due to their cytocompatibility, biodegradability, anticoagulant, and hemostatic properties [322,327]. Chitosan stimulates hemostasis, accelerating the blood clotting process and tissue regeneration [325]. Chitosan-based materials also exhibit antimicrobial activities, as discussed above, thereby protecting wounds from infection during tissue healing [328]. Physical chitosan-based hydrogel assemblies can absorb a high content of water and biological fluids. Wet wound dressings can be added to the wounds to provide moisture and prevent tissue dehydration. Additionally, the normal healing process can be accelerated by absorbing exudates from the wounds [324].

Tamer and coworkers developed chitosan/hyaluronic acid wound dressing (films) containing with glutathione (an anti-inflammatory and antioxidant agent). The material accelerated skin wound healing in rats after 18 days. The material supported the formation of connective tissue and collagen, promoting healing. However, the complete healing was only obtained by incorporating the anti-inflammatory glutathione in the material [322].

The disadvantages of polysaccharide-based wound dressings include their brittle mechanical properties characterized by high tensile strength and low elasticity. Additionally, the most used polysaccharide (chitosan) is insoluble in neutral or alkaline environments, which is a problem in wound dressing applications. These issues can be overcome by associating polysaccharides with synthetic and semi-synthetic polymers, such as poly(vinyl alcohol) [329], polyethylene glycol fumarate [330], silver nanoparticles [327], inorganic salts (containing Mg^2+^, Ca^2+^, or Ba^2+^ ions) [331] and other additives. In addition to improving mechanical properties, reinforcing the polymer matrices with additives, can also improve the antibacterial activity [327].

The principal criticism about this topic is the wound dressing definition. Wound dressings materials should not form strong interactions with the tissue surfaces, because they need to be replaced without causing pain to the patients. The definition of wound dressing materials is often incorrectly presented in the literature. Scaffolds are frequently referred to as wound dressings; however, the main function of scaffolds is to accelerate the wound-healing process [322,325]. Scaffolds are not wound dressings because these materials interact with damaged tissues during the healing process, and cannot be removed from the tissue without causing pain or potentially damaging the wound site. Moreover, studies reporting physical materials as wound dressings rarely present clinical outcomes.

## 5. Summary and Perspective

Polysaccharide-based 3D porous scaffolds, coatings, and wound dressings have advantages over synthetic and semisynthetic materials. Glycosaminoglycans, marine polysaccharides, and cellulose have been used in biomedical applications because they can mimic the ECM composition and function, stabilizing growth factors (GFs). Polysaccharide assemblies can be engineered following many strategies, including electrospinning, polyelectrolyte complexation, ionotropic gelation, layer-by-layer assembly of PEMs, solvent evaporation, gelation of polymer solutions, freezing–thawing, and 3D bioprinting approaches. The biological features of polysaccharide assemblies are significantly enhanced when associated with GFs. Polysaccharide-based scaffolds and coatings for wound healing and tissue repair must be formulated to stabilize and deliver GFs. These materials must stabilize the GFs against degradation and control GF release and presentation to avoid side-effects. Polysaccharide-based assemblies are excellent candidate materials for GF delivery, as they are capable of binding and stabilizing GFs and releasing GFs with tunable kinetics or in response to the local biochemical environment. Furthermore, polysaccharides can be processed into tissue scaffolds with excellent properties for supporting cell growth, migration, and organization. These scaffold materials can be designed to degrade at controlled rates, and they offer additional biological activities that can promote tissue healing.

Polysaccharide-based wound dressings have also received significant attention because chitosan and its derivatives have bactericidal properties, and because polyanionic polysaccharides (especially those that are sulfated) have anti-adhesive properties. However, chitosan has aqueous insolubility requiring dilute acid solutions for complete solubilization. The remaining acid content in the wound dressings can influence the antimicrobial traits. Physical assemblies can have weak mechanical properties and low durability depending on the strategy used to yield the materials and polymers that comprise the assemblies.

This review presented current and innovative results concerning the polysaccharide-based systems. The material durability can also be achieved by associating polymers with high molar masses and ionizable groups in their structures, controlling the experimental condition (pH, temperature, and ionic strength) used to process the materials. Surface coatings based on sulfated glycosaminoglycans and polyphenolic tannins have outstanding blood compatibility, while porous hydrogels based on chitosan support mammalian cell proliferation and growth. Several drug delivery systems (DDSs) are projected for GFs delivery. These can increase GF stability, promoting their sustained release. Physical materials are engineered without the use of chemical crosslinking agents often used to provide durable polymeric materials. These chemistries can reduce biodegradability and cytocompatibility, preventing the use of polysaccharide-based materials in biomedical applications. These disadvantages can be overcome by designing physical materials, using polyelectrolytes with opposite charges in aqueous media.

## Figures and Tables

**Figure 1 pharmaceutics-13-00621-f001:**
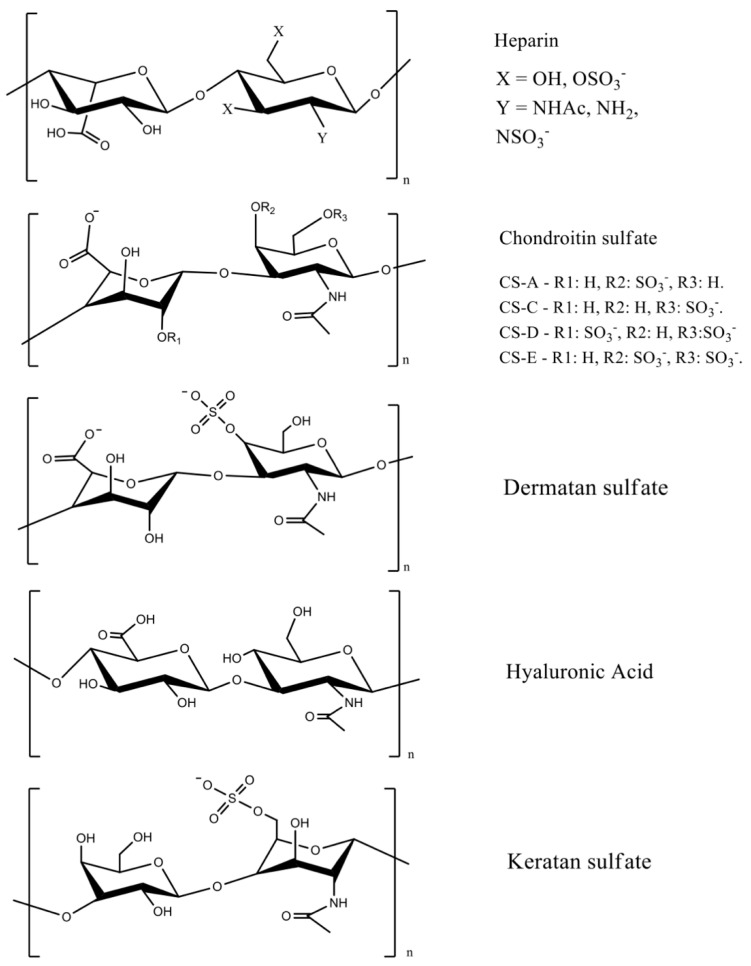
GAG chemical structures. Adapted with permission from [7,15]; published by Elsevier 2019 and Wiley, 2020. CS = chondroitin sulfate, and the letters A, B, C, D, and E represent different types of chondroitin sulfates.

**Figure 2 pharmaceutics-13-00621-f002:**
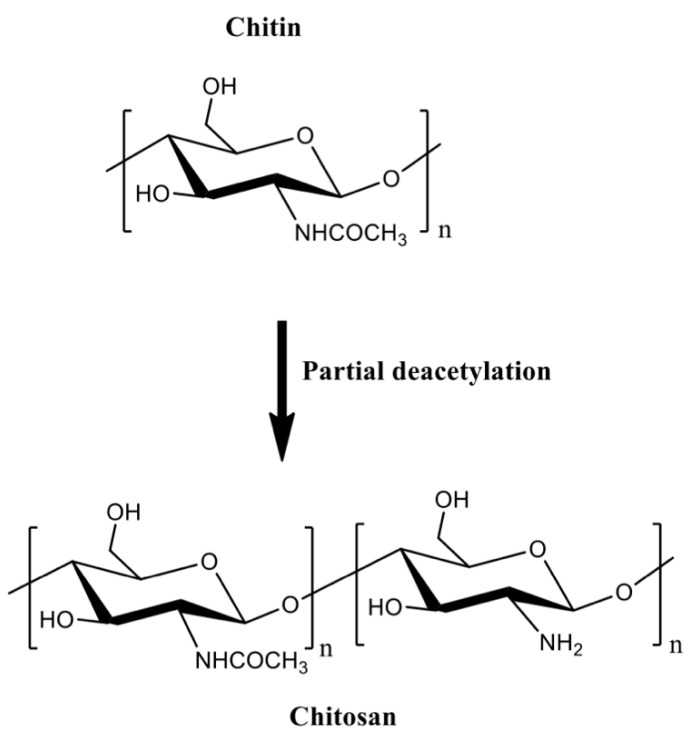
The partial chitin deacetylation produces chitosans with deacetylation degrees higher than 50%.

**Figure 3 pharmaceutics-13-00621-f003:**
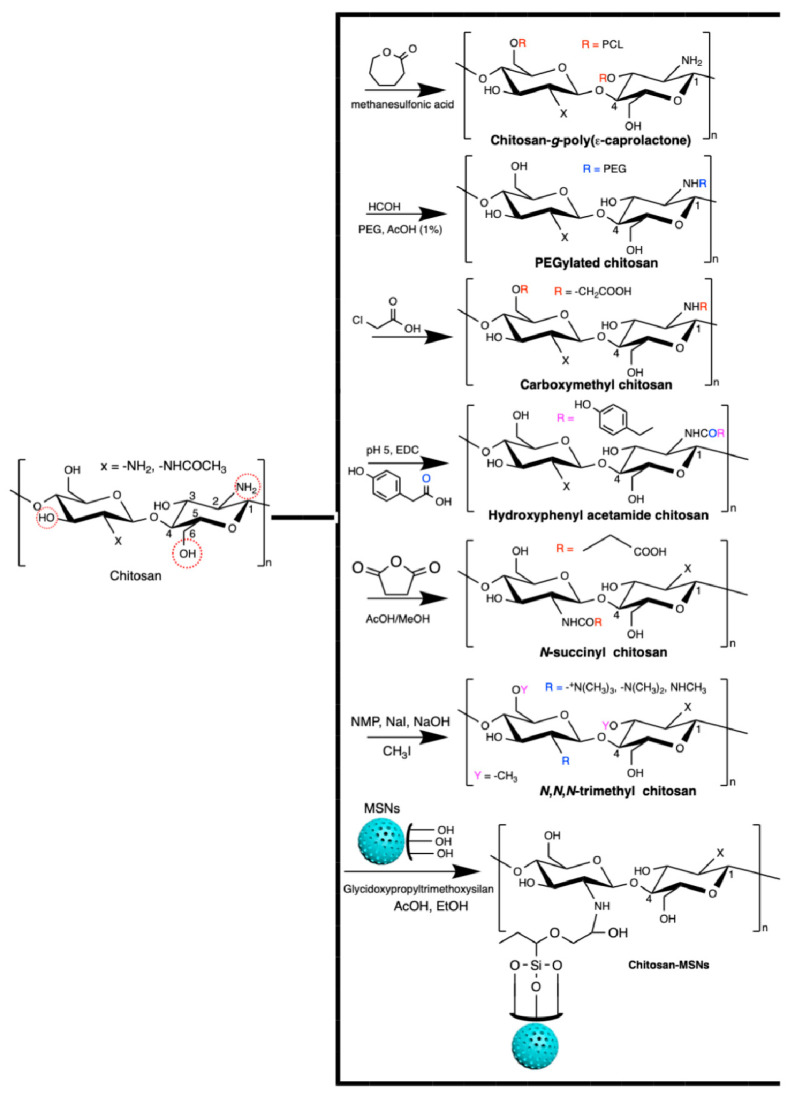
Chitosan-based materials used in biomedical-engineering applications.

**Figure 4 pharmaceutics-13-00621-f004:**
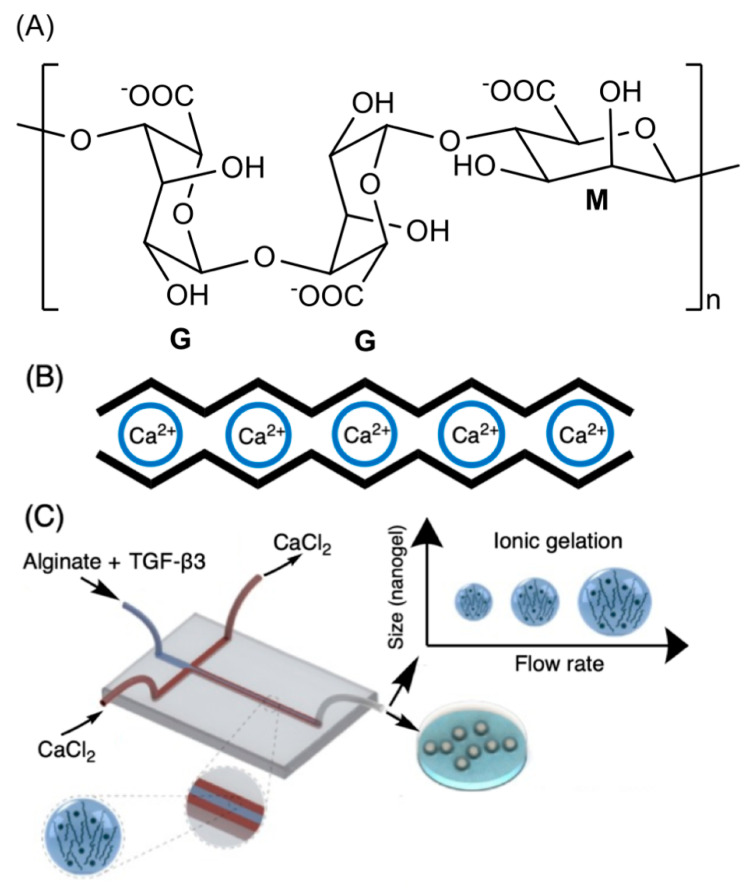
Chemical alginate structure (**A**), well-established egg-box gelation of alginate with calcium ions (**B**), and schematic illustration of a microfluidic device for hydrodynamic flow-focusing consisting of one inlet for focusing (core) flow and two separate inlets for the sheath (side) flows. (**C**) Adapted with permission from [88]; published by Elsevier, 2020.

**Figure 5 pharmaceutics-13-00621-f005:**
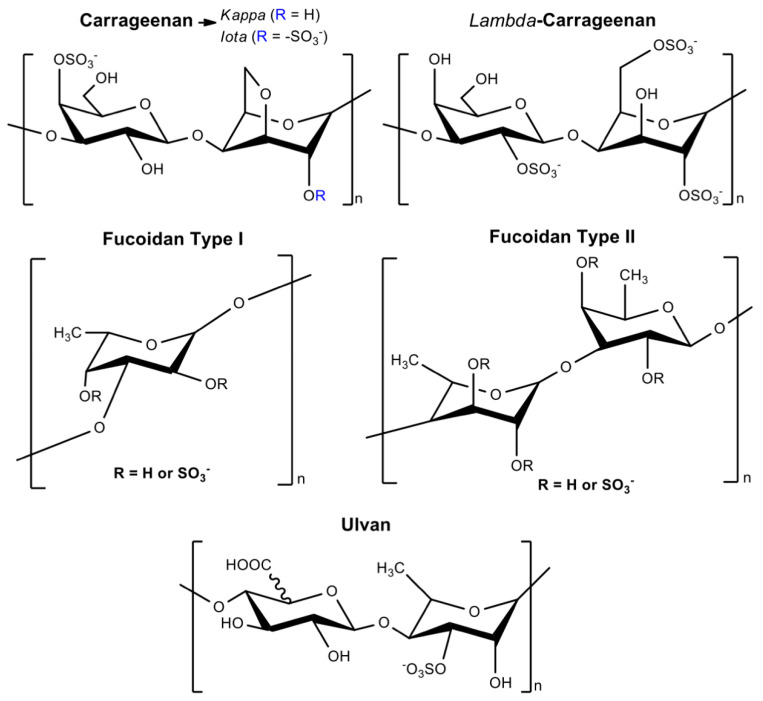
Chemical structures of sulfated marine polysaccharides. Adapted with permission from [7,101], published by Elsevier, 2019 and 2020.

**Figure 6 pharmaceutics-13-00621-f006:**
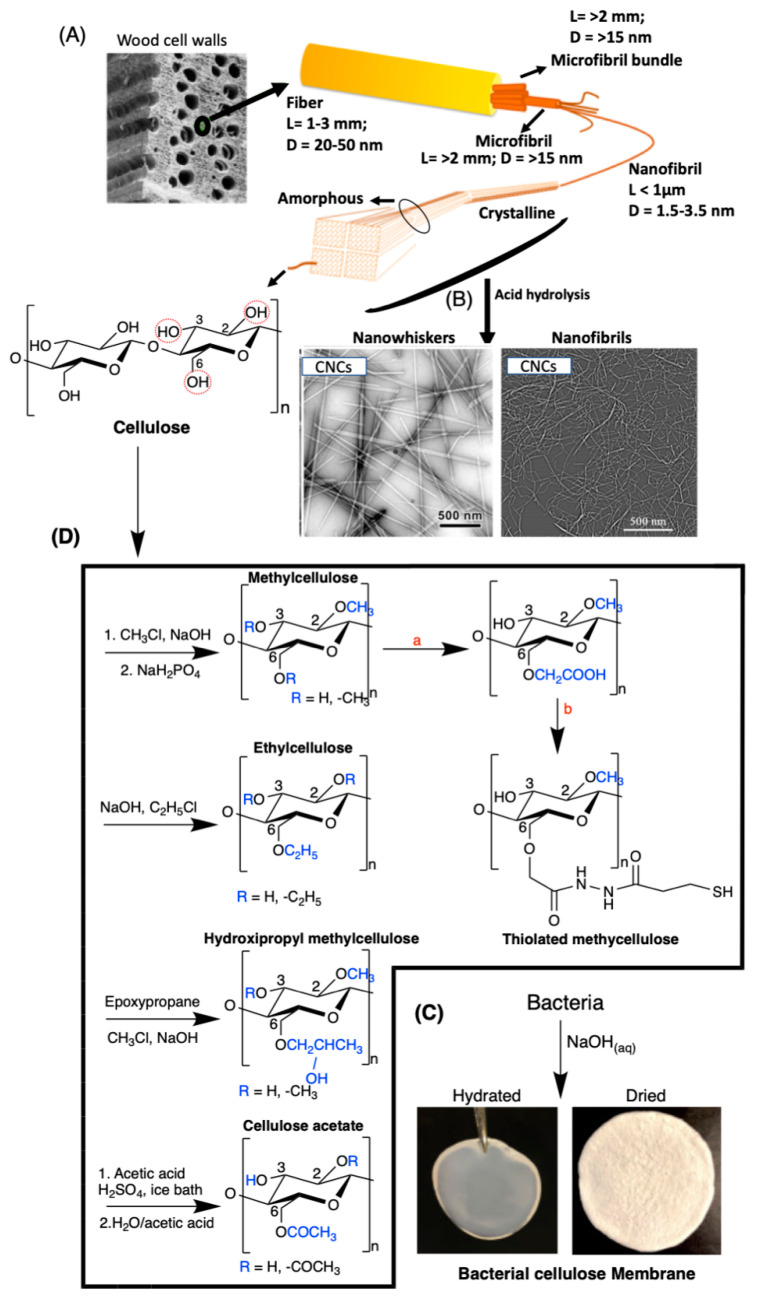
(**A**) Adapted with permission from References [122,123]; Published by Elsevier, 2020 and IntechOpen, 2015, respectively. (**B**) Adapted with permission from [123,124]; published by IntechOpen, 2015 and American Chemical Society, 2017, respectively. (**C**) Adapted with permission from Reference [118]; Published by Elsevier, 2020. (**D**) Adapted with permission from References [125,126,127]; published by Elsevier, 2019 and 2020 and American Chemical Society, 2012, respectively. Data: a = 3 M bromoacetic acid, 1 M NaOH, 3 h, 4 °C. b (i) 1-ethyl-3-[3-(dimethylamino)propyl]-carbodiimide, 3,3′-dithiobis (propionic dihydrazide), pH 4.5, 2 h, room temperature; (ii) dithiothreitol, pH 8.5, 24 h, room temperature. L = length, D = diameter, CNCs = cellulose nanocrystals.

**Figure 7 pharmaceutics-13-00621-f007:**
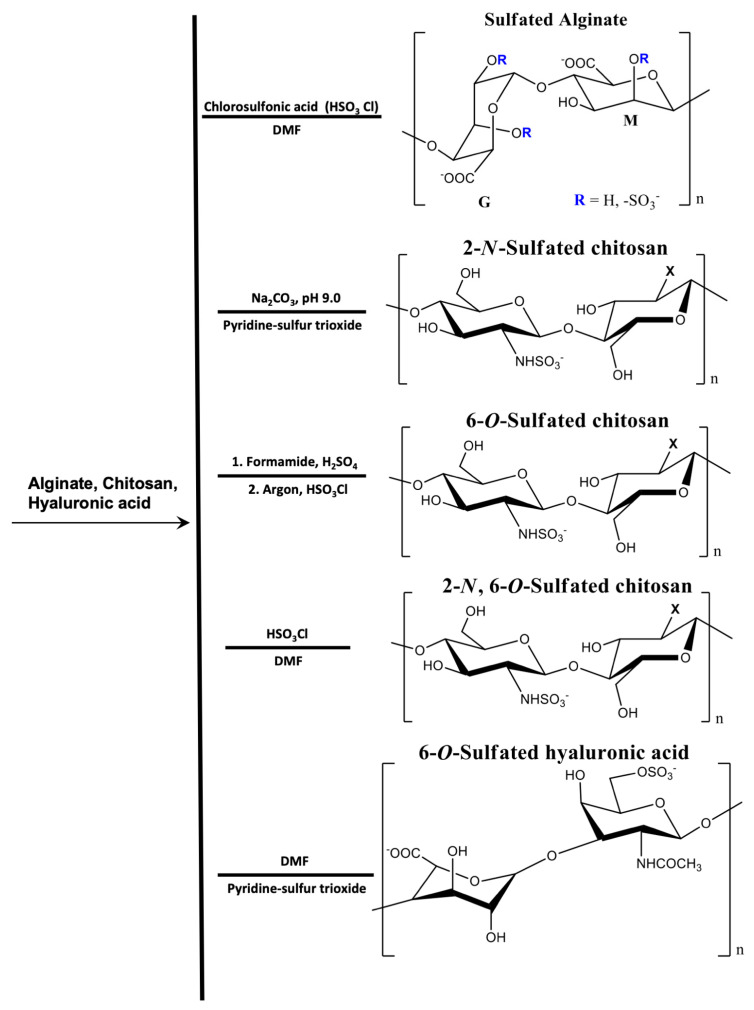
Chemical structures of artificially sulfated polysaccharides obtained from alginate, chitosan, and hyaluronic acid.

**Figure 8 pharmaceutics-13-00621-f008:**
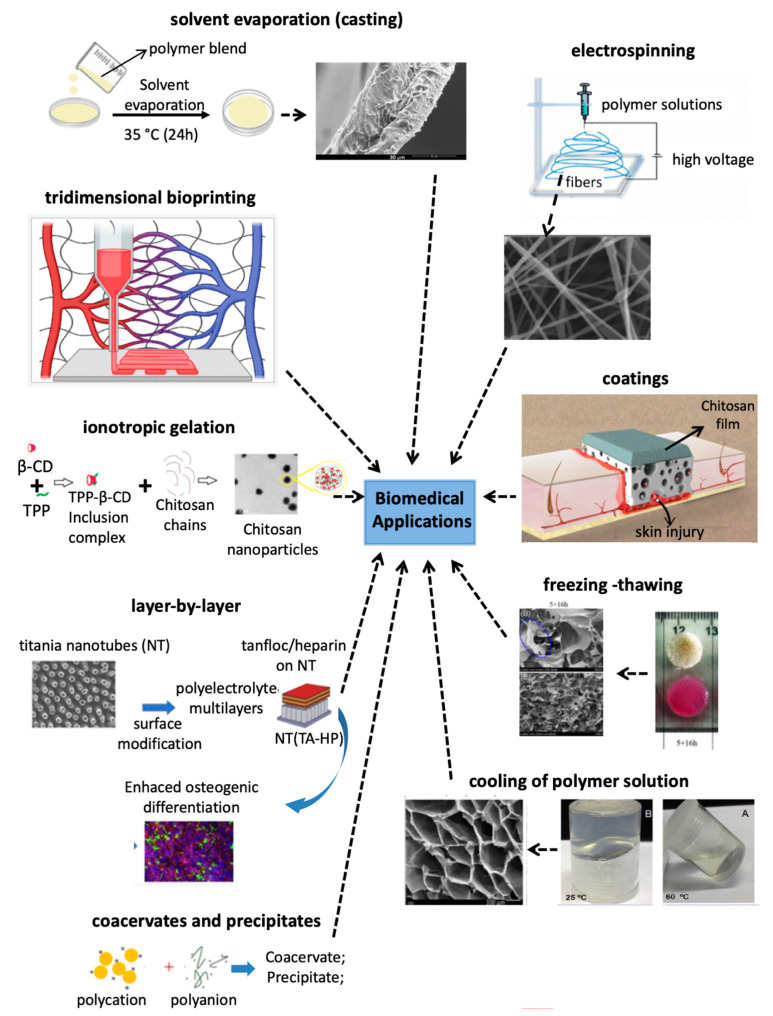
Principal strategies used to create polysaccharide-based materials by physical association for biomedical applications. Adapted with permission from References [139,140,141,142,143,144,145]; published by Elsevier, 2019, 2020, 2018, 2020, 2020, 2010, and 2020, respectively. Adapted with permission from Reference [146]; published by American Chemical Society, 2020. Adapted from the References [147,148]; published by Wiley, 2019 and 2020. TA = polyphenolic tannin derivative, commercially called Tanfloc; HP = heparin; β-CD = β-cyclodextrin; and TPP = tripolyphosphate.

**Figure 9 pharmaceutics-13-00621-f009:**
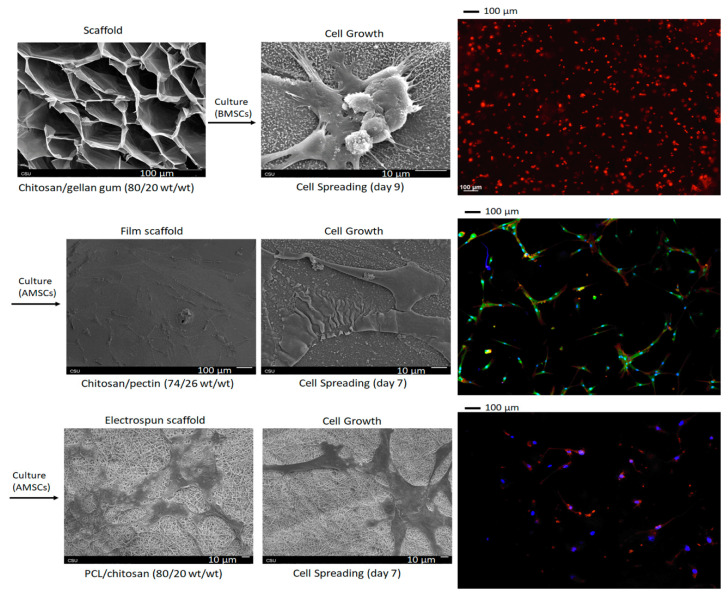
SEM and fluorescence images of mammalian cells seeded on polysaccharide-based scaffolds. (PCL = poly(ε-caprolactone), BMSCs = bone-marrow-derived mesenchymal stem cells and AMSCs = adipose-derived mesenchymal stem cells.) Reprinted with permission from References [138,145]; published by Elsevier, 2018 and 2020, respectively. Reprinted with permission from Reference [206]; published by SciELO, 2019.

**Figure 10 pharmaceutics-13-00621-f010:**
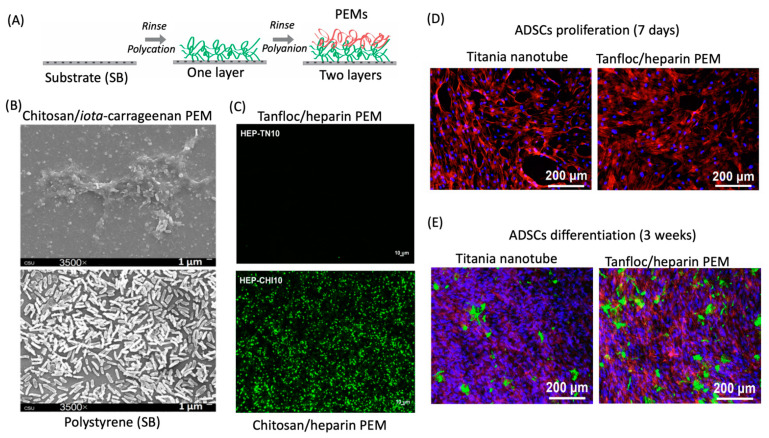
Schematic of PEM assembly onto a oxidized solid substrate (**A**) (adapted with permission from Reference [238]; published by Royal Society of Chemistry, 2019. SEM images of chitosan/*iota*-carrageenan PEM (15 layers) and native polystyrene (solid substrate, SB) seeded with *P. aeruginosa* after 6 h of incubation (**B**). Adapted with permission from Reference [37]; published by Elsevier, 2020. Representative fluorescence images of adhered platelets stained with calcein-AM on Tanfloc/heparin and chitosan/heparin PEMs (10 layers) (**C**). Adapted with permission from Reference [154]; published by Elsevier, 2020. Fluorescence images of adipose-derived mesenchymal stem cells (AMSCs) stained with DAPI (blue) and rhodamine-phalloidin (red) imaged on the titania nanotubes and Tanfloc/heparin PEM (5 layers) after 7 days of culture (**D**). Adapted with permission from Reference [140]; published by Elsevier, 2021. Representative immunofluorescence microscopy images of ADSCs after 3 weeks of induced osteogenesis on titania nanotubes and Tanfloc/heparin PEM (5 layers), in which the green stain indicates osteocalcin (**E**). Adapted with permission from Reference [140]; published by Elsevier, 2021.

**Figure 11 pharmaceutics-13-00621-f011:**
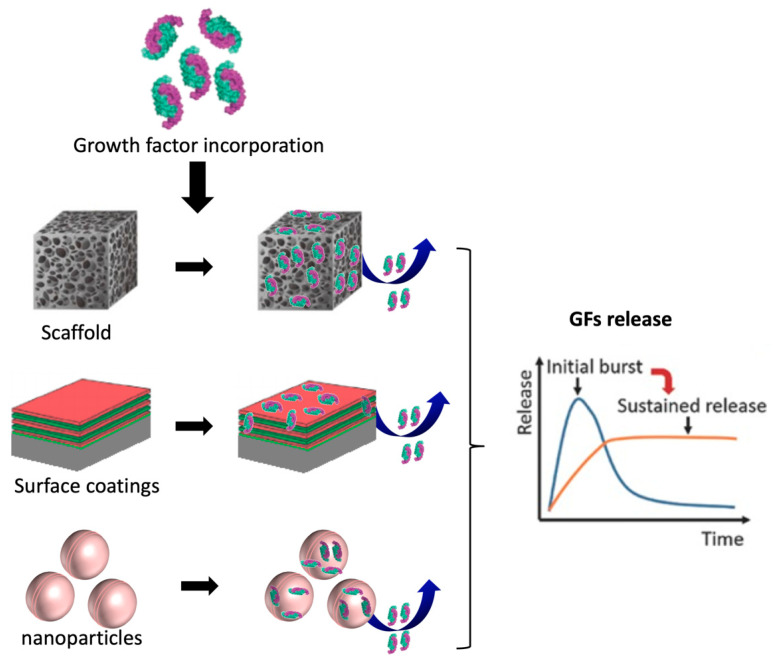
Materials (scaffolds, surface coatings, and nanoparticles) as DDSs used for efficient growth factor delivery in biomedical-engineering applications by promoting a sustained release. Adapted with permission from References [15,243,268]; published by Wiley, 2020 and 2016, and MDPI, 2011, respectively.

**Table 1 pharmaceutics-13-00621-t001:** Physical assemblies (scaffolds) based on polysaccharides.

Scaffolds	Approach	Cells	Tissue	References
GG/Manuka honey	Ionotropic gelation (Ca(II)/Mg(II))	MSCs	Cartilage	[205]
CHT/GG	Gelation/freeze-drying	BMSCs	Bone	[145]
CHT/PT	Solvent evaporation	AMSCs	Skin	[138]
CHT/GE/HA/PEO	Electrospinning	HDF	Skin	[195]
CHT/GE	Freezing–thawing	BMSCs	Bone	[54]
CHT/HA/PEO	Electrospinning/layer-by-layer; solvent evaporation	MCs		[212]
CHT/CS; CHT/ALG	Precipitation/freeze-drying	PC3	Epithelial	[208,213]
CHT/ALG	Precipitation/freeze-drying/Ca(II)	Fibroblasts	Cartilage	[209]
CHT/ALG; CHT/PT	Solvent evaporation	HDF		[214]
ALG/PEO/pluronic^®^ F127	Single nozzle electrospinning	MCs		[215]
CHT/HA	Solvent evaporation/freeze-drying	GBs		[216]
ALG/XG/*κ*CA/CHT/GE	3D printing layer-by-layer	C2C12		[217]
CHT*/κ*CA	Precipitation/freeze-drying	-		[218]
CHT/ulvan	Precipitation/freeze-drying	MC3T3E-1	Bone	[219]
CHT/silk fibroin	Precipitation/freeze-drying	MSCs	Bone	[220]
Cationic tannin/ALG	Precipitation/freeze-drying	MC3T3-E1	Bone	[221]
ε-Polylysine/HP-PO	Gelation/freeze-drying	ECCs	Skin	[222]
CHT/PCL	Electrospinning	MSCs, PC12	Skin, neural	[206,223,224]
CHT/γ-PLGA	Precipitation/freeze-drying	Fibroblasts	Skin	[225]
ALG/HA/PEI/PVA	Precipitation/freeze-drying/Ca(II)	Schwann		[226]
KG/PVA	Freezing–thawing/Ca(II)	BMSCs	Bone	[227]
CHT/PVA	Freezing–thawing	C6 glioma	Neural	[228]
HA/ALG/PVA/PEG	Solvent evaporation	Fibroblasts		[229]
ALG	Ionotropic gelation/solvent evaporation	-	Bone	[230]

Samples: ALG = alginate; *κ*CA = *κ*-carrageenan; CHT = chitosan; CS = chondroitin sulfate; GE = gelatin; GG = gellan gum; KG = karaya gum; HA = hyaluronic acid; HP = heparin; PCL = poly(ε-caprolactone); PEG = poly(ethylene glycol); PEI = polyethyleneimine; γ-PLGA = γ-polyglutamic acid; PO = poloxamer; PT = pectin; PVA = poly(vinyl alcohol); PEO = polyethylene oxide; XG = xanthan gum. Cells: AMSCs = adipose-derived mesenchymal stem cells; BMSCs = bone-marrow derived mesenchymal stem cells; C2C12 = mouse myoblasts cells; ECCs = uterine endometrial carcinoma cell; GBs = glioblastoma cells; HDF = human dermal fibroblasts; MCs = mammalian cells; MC3T3-E1 = mouse pre-osteoblastic cells; PC12 = neuroblastic and eosinophilic cells; PC3 = prostate cancerous cells.

**Table 2 pharmaceutics-13-00621-t002:** Principal tissues and healing processes associated with the GFs associated with physical polysaccharide-based materials.

Biological Process	GFs Involved	References
Bone healing, differentiation, and survival of osteoblasts and osteoclasts	FGF-2; PDGF; BMP-2 and -7; IL-1 and -6; VEGF; IL-3	[241,242,243]
Cartilage healing and chondrocyte differentiation	TGF-β1 and -β3; IL-1; BMP-2 and -7; FGF-2, -3, and -18; CTGF; PDGF; IGF-1; NGF; IL-1	[244,245,246,247]
Blood-vessel formation (angiogenesis)	VEGF; PlGF; TGF-β; HGF; IGF-1; FGF-2; pleiotrophin; PDGF; erythropoietin; angiopoietin	[240]
Nerve survival, differentiation, maturation, and regeneration	NGF; brain derived neurotrophic factor (BDNF); ciliary neurotrophic factor (CNTF); neurotrophin-3 and -4/5; epidermal growth factor (EGF); glial-cell-derived neurotrophic factor; IL-6; PDGF; FGF-1 and -2; IGF; VEGF	[248,249,250]
Skin wound healing		
Liver regeneration, hepatocyte maintenance, and stellate cell signaling	HGF; TGF-β1 and -α; EGF; HB-EGF; IL-6; FGF-1 and -2; insulin; PDGF; angiopoietin-1 and -2; VEGF; IGF; BMP-7	[251,252]

## Data Availability

Not applicable.

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
