# Peer review of "Polysaccharide-Based Materials Created by Physical Processes: From Preparation to Biomedical Applications"

_pharmaceutics, 2021, doi:10.3390/pharmaceutics13050621_

Round 1

Reviewer 1 Report

This article presents a review on polysaccharide-based materials created by physical processes. As the authors claim, this type of materials is has received considerable attention for tissue engineering applications. Therefore, this review also focuses on materials used for growth factor (GFs) delivery, tissue engineering, scaffolds, antimicrobial coatings, and wound dressings. Therefore, I recommend the publication of this manuscript after the following minor revisions.

  1. In Figure 1, the structure of heparin is incorrect. A bond crossing the pyranose ring should be placed at behind of the ring. The same mistakes are appeared in Figures 4, 5, and 7.
  2. Regarding carboxymethyl chitosan, there are three types, that is, O-, N-, and N,O-carboxymethyl chitosans. The explanation on types of carboxymethyl chitosans should be added on page 6.

Author Response

Reviewer #1

This article presents a review on polysaccharide-based materials created by physical processes. As the authors claim, this type of materials is has received considerable attention for tissue engineering applications. Therefore, this review also focuses on materials used for growth factor (GFs) delivery, tissue engineering, scaffolds, antimicrobial coatings, and wound dressings. Therefore, I recommend the publication of this manuscript after the following minor revisions.

Thank you very much, Reviewer #1.

  1. In Figure 1, the structure of heparin is incorrect. A bond crossing the pyranose ring should be placed at behind of the ring. The same mistakes are appeared in Figures 4, 5, and 7.

Thank you very much, Reviewer #1. We corrected all Figures.

  1. Regarding carboxymethyl chitosan, there are three types, that is, O-, N-, and N,O-carboxymethyl chitosans. The explanation on types of carboxymethyl chitosans should be added on page 6.

Thank you very much, Reviewer #1. We reported about the other kinds of carboxymethyl chitosan in the revised manuscript.

Reviewer 2 Report

The authors present a review on the use of PS-based materials for biomedical applications, limiting their preparation to physical methods i.e. without covalent crosslinking. The 31-page review (excluding references) is supported by 11 figures, 2 tables and over 300 bibliographic references. The manuscript is divided into 4 parts: (1) a general introduction, (2) the presentation of the main polysaccharides referenced, (3) the main strategies to prepare the materials, (4) the applications of materials in the biomedical field with a particular focus on the controlled release of growth factors.

A summary is necessary.

The authors seem to limit their work to tissue engineering (Line 2 of the abstract, line 59) while controlled release, which cannot be included in tissue engineering, is largely developed in the following. Briefly outline tissue engineering is necessary. (Note, tissue engineering is widely considered to be a part of nanomedicine, the term of which is never cited).

Correct the first sentence of the introduction because the hydroxyl groups of PS are not ionizable in water.

Part (2) which comprises 12 pages should be reduced drastically and rewritten in a more direct and clear manner. Indeed, the reviews in this field are numerous, varied and regularly updated. The informed reader does not learn anything that will allow a better understanding of parts (3) and (4). In addition, dextran and pullulan derivatives should appear there.

Line 67: correct “stabile”. Line 79: insert a space between “acid” and “or”. Line 86-87; the sentence "Heparin ... solutions" doesn't make sense. In Figure 1: the structure of heparin needs to be corrected; specify the meaning of n1-n4. Correct “chtitin” in figure 2. Correct the structures of figures 4 and 5 (badly positioned chemical links). Line 103: specify that these values concern the pKa of the carboxylic groups. Line 165: replace “biopolymers” by “polymers”. Lines 206-207: the sentence “Alginates… groups » (obvious) could be deleted. Line 240: remove "Fucus vesiculosus" or add other species. Lines 245-246: the sulphate composition of fucoidans varies from 1 to 3 for 2 units: modify accordingly. Lines 250-252: the general sentence “Sulfated… applications." is in the wrong place. Lines 257-259: The number of cellulose-producing plant species is so high that the proposed references do not make sense. The only reference 122 is sufficient.

In general, the biodegradability of PS is mentioned as an advantage for their use in the biomedical field. Please clarify this notion.

Parts (3) and (4) are interesting, well written, clear and informative. Authors should refocus their manuscript on these parts. Paragraph 4.3 needs to be rearranged, the first paragraph (lines 756-770) is clearly not in the right place (suggestion: insert on line 837). The paragraph 4.4 ("wound healing") could be expanded. A paragraph on clinical outcomes could be added (and why so few outcomes as compared to the plethora of studies ?).

In the end, the manuscript must be widely modified and improved before considering publication.

Author Response

Reviewer #2

The authors present a review on the use of PS-based materials for biomedical applications, limiting their preparation to physical methods i.e. without covalent crosslinking. The 31-page review (excluding references) is supported by 11 figures, 2 tables and over 300 bibliographic references. The manuscript is divided into 4 parts: (1) a general introduction, (2) the presentation of the main polysaccharides referenced, (3) the main strategies to prepare the materials, (4) the applications of materials in the biomedical field with a particular focus on the controlled release of growth factors.

Thank you very much, Reviewer #2.

A summary is necessary.

Thank you very much, Reviewer #2. The manuscript template does not indicate an appropriate place to add a summary. We will investigate if it is possible to add the summary.

The authors seem to limit their work to tissue engineering (Line 2 of the abstract, line 59) while controlled release, which cannot be included in tissue engineering, is largely developed in the following. Briefly outline tissue engineering is necessary. (Note, tissue engineering is widely considered to be a part of nanomedicine, the term of which is never cited).

Thank you very much, Reviewer #2. We made some mistakes. In some places, we have changed the wording to expand the scope from “tissue engineering” to more general “biomedical applications.” This occurred in lines 18, 67, 268, 370, and others. 

The controlled release of growth factors can be included in tissue engineering applications. We do not mention the term nanomedicine because the drug delivery systems used for growth factor delivery are porous hydrogels, microparticles, nanoparticles, films, coatings, fibers, and others. Therefore, nanosystems are only a subset of those used for efficient grow factor delivery.

We removed the word “tissue engineering” from the abstract. Overall, we report polysaccharides and the principal strategies to create physical materials for biomedical applications.

Correct the first sentence of the introduction because the hydroxyl groups of PS are not ionizable in water.

Thank you very much, Reviewer #2. The sentence is correct. We said that polysaccharides have ionizable sites as well as sites available to interact through intermolecular interactions (e.g., H-bonds, dipole-dipole forces, etc.).

Part (2) which comprises 12 pages should be reduced drastically and rewritten in a more direct and clear manner. Indeed, the reviews in this field are numerous, varied and regularly updated. The informed reader does not learn anything that will allow a better understanding of parts (3) and (4). In addition, dextran and pullulan derivatives should appear there.

Thank you very much, Reviewer #2. We have reported in this section more than 25 types of polysaccharides, including derivatives. Also, we present seven large Figures in part (2). Because of this, part (2) has about 12 pages. In our opinion, this part is crucial to the manuscript because we present the polysaccharides, commenting about extraction and production methods, principal properties, and some advantages and disadvantages for biomedical applications.

In this manuscript, we highlighted the principal polysaccharides with ionizable sites in their chains. Many other (charged and non-charged) polysaccharides were not reported, including gums, pectin, pullulan, dextran, etc. We focus on glycosaminoglycans, marine polysaccharides, cellulose, and derivatives. Our principal focus is on charged polyelectrolytes (polyanionic and polycationic polymers) because these can mainly interact through electrostatic interactions, forming durable assemblies (physical materials) for biomedical applications. Also, we focus on cellulose (a non-charged polysaccharide) because it is the most abundant polysaccharide in the world, providing nanocrystalline structures that improve the mechanical properties of polysaccharide-based materials and bacterial cellulose are attracting significant attention in biomedical applications.

We added this information in the revised manuscript. Also, we linked to section 2 to section 3.

The new text added to the manuscript is in red.

Line 67: correct “stabile”. It was corrected. Line 79: insert a space between “acid” and “or” It was corrected. Line 86-87; the sentence "Heparin ... solutions" doesn't make sense. We removed this sentence. In Figure 1: the structure of heparin needs to be corrected. It was corrected. Specify the meaning of n1-n4. We corrected this. Correct “chtitin” in figure 2. It was corrected. Correct the structures of figures 4 and 5 (badly positioned chemical links). It was corrected. Line 103: specify that these values concern the pKa of the carboxylic groups. It was corrected. Line 165: replace “biopolymers” by “polymers”. It was corrected. Lines 206-207: the sentence “Alginates… groups » (obvious) could be deleted. The sentence was removed. Line 240: remove "Fucus vesiculosus" or add other species. It was removed. Lines 245-246: the sulphate composition of fucoidans varies from 1 to 3 for 2 units: modify accordingly. Sorry, we did not find this sentence in the manuscript. Lines 250-252: the general sentence “Sulfated… applications." is in the wrong place. We added this sentence in the first paragraph in section 2.4. Lines 257-259: The number of cellulose-producing plant species is so high that the proposed references do not make sense. The only reference 122 is sufficient. It was corrected.

Thank you very much, Reviewer #2. We revised the manuscript and Figures.

In general, the biodegradability of PS is mentioned as an advantage for their use in the biomedical field. Please clarify this notion.

Thank you very much, Reviewer #2. We have added the sentences below in the manuscript. Please, see Section 4.3.1.

Controlled protein delivery is often achieved through the use of biodegradable polymers. Tuning the rate of degradation is one means of controlling the drug release rate. Many polysaccharides are biodegradable by enzymes in mammalian tissues. Their degradation products are generally non-toxic saccharides, which are expected to have a minimal burden on metabolic processes.

Parts (3) and (4) are interesting, well written, clear and informative. Authors should refocus their manuscript on these parts. Paragraph 4.3 needs to be rearranged, the first paragraph (lines 756-770) is clearly not in the right place (suggestion: insert on line 837).

Thank you very much, Reviewer #2. We corrected this.

The paragraph 4.4 ("wound healing") could be expanded. A paragraph on clinical outcomes could be added (and why so few outcomes as compared to the plethora of studies?).

Thank you very much, Reviewer #2. We did not find papers that describe the clinical use of polysaccharide-based materials as appropriate wound dressings. There are many materials used as scaffolds for skin repair. However, wound dressing materials should not interact with the native tissue. The main functions of wound dressings include non-toxicity and non-allergenic, protect the wound against pathogens, and provide absorption of wound exudates. Polysaccharide-based materials are often wrongly reported in the literature, as would dressings. These materials are scaffolds, which primarily function to accelerate the wound healing process. They cannot be called wound dressings because they interact with the native tissues and can not be replaced without causing pain or damage to the wound site. Also, studies reporting physical materials as wound dressings rarely present clinical outcomes.

We added this information in the revised manuscript.

In the end, the manuscript must be widely modified and improved before considering publication.

Thank you very much, Reviewer #2. We improved the manuscript following your suggestions.

Reviewer 3 Report

The manuscript submitted by Paulo R. Souza et. al. reviewed the principal polysaccharides and strategies creating biomedical polysaccharide-based assemblies. The author made a comprehensive review for the strategies of polysaccharides in biomedical. But this MS needs major revision and the author reply comments properly before accepted by Pharmaceutics. The comments and questions are as follows:

Firstly, Line158, “Chitosan is the only commercially available cationic polysaccharide.” Is this expression correctly? Please make a proper description.

Line 281, please describe the disadvantages of the extraction from algae and animals clearly.

Then, in the section 3. does different physical process for the same polysaccharides produce different structure or property? Also, for different usage, are there any demand for the processed polysaccharides structure?  

Comparing to the original polysaccharides or the polysaccharides-based materials created by chemical process, what is the obviously advantage or disadvantage of polysaccharide-based material created by physical process?

Lastly, the manuscript reviewed the polysaccharides in biomedical application detailly. But, any the whole new opinion or creative thought the authors want to bring forward? So please describe it in the summary part.

Totally, this manuscript can be published after some minor revisions and giving proper answers for those question.

Author Response

Reviewer #3

The manuscript submitted by Paulo R. Souza et. al. reviewed the principal polysaccharides and strategies creating biomedical polysaccharide-based assemblies. The author made a comprehensive review for the strategies of polysaccharides in biomedical. But this MS needs major revision and the author reply comments properly before accepted by Pharmaceutics. The comments and questions are as follows:

Thank you very much, Reviewer #3.

Firstly, Line158, “Chitosan is the only commercially available cationic polysaccharide.” Is this expression correctly? Please make a proper description.

We removed this sentence from the revised manuscript because some bacteria species can synthesize chitosan. Chitosan is also a natural polysaccharide.

Line 281, please describe the disadvantages of the extraction from algae and animals clearly.

Thank you very much, Reviewer #3. We added this information was added in the revised manuscript.

Then, in the section 3. does different physical process for the same polysaccharides produce different structure or property? Yes. Also, for different usage, are there any demand for the processed polysaccharides structure? Yes.

Thank you very much, Reviewer #3. We clarified this in the revised manuscript.

Comparing to the original polysaccharides or the polysaccharides-based materials created by chemical process, what is the obviously advantage or disadvantage of polysaccharide-based material created by physical process?

Thank you very much, Reviewer #3. We clarified this in the revised manuscript.

Lastly, the manuscript reviewed the polysaccharides in biomedical application detailly. But, any the whole new opinion or creative thought the authors want to bring forward? So please describe it in the summary part.

Thank you very much, Reviewer #3. We clarified this in the revised manuscript.

Totally, this manuscript can be published after some minor revisions and giving proper answers for those question.

Thank you very much, Reviewer #3.

Round 2

Reviewer 2 Report

Following the modifications and corrections made by the authors, the manuscript is clearer and could be published subject to some minor corrections.

Once again, the first sentence of the introduction must be modified because it suggests that all the polysaccharides carry ionizable groups, that is to say organic functions with a negative (carboxylate, sulfate, etc.) or positive ( ammonium) charge depending on the pH range of their solution, when they are soluble in water, which is not the case for all polysaccharides (eg cellulose, etc.).

There remains a structural error in fucoidan structure in Figure 5: poly L-fucose and not D-fucose; and the sentence in lines 294-295 of v2 ("Fucoidan resembles the heparin structure because it has three sulfate sites per disaccharide") must be modified: not all fucoidans carry 3 sulfates for 2 units, the variability is greater according to plant (algae) or animal (tunicates) species: between 1 and 3 sulfates per disaccharide.

Author Response

I have attached the response.

Thank you very much.
